# Current Catalyst Technology of Selective Catalytic Reduction (SCR) for NO$_x$ Removal in South Korea

**Hyo-Sik Kim [1,2], Saravanan Kasipandi [2], Jihyeon Kim [1], Suk-Hwan Kang [1], Jin-Ho Kim [1], Jae-Hong Ryu [1,*] and Jong-Wook Bae [2,*]**

[1] Plant Engineering Center, Institute for Advances Engineering (IAE), 175-28, 51, Goanro, Baegam-myeon, Cheoin-gu, Yongin-si 449-863, Gyeonggi, Korea; hyosgogo@iae.re.kr (H.-S.K.); kimjyun@iae.re.kr (J.K.); shkang@iae.re.kr (S.-H.K.); jinho@iae.re.kr (J.-H.K.)

[2] School of Chemical Engineering, Sungkyunkwan University (SKKU), 2066 Seobu-ro, Jangan-gu, Suwon 16419, Gyeonggi-do, Korea; saravananspkc@gmail.com

[*] Correspondence: jhryu@iae.re.kr (J.-H.R.); finejw@skku.edu (J.-W.B.); Tel.: +82-31-330-7882 (J.-H.R.); +82-31-290-7347 (J.-W.B.); Fax: +82-31-330-7850 (J.-H.R.); +82-31-290-7272 (J.-W.B.)

**Abstract:** Recently, air pollution has worsened throughout the world, and as regulations on nitrogen oxides (NO$_x$) are gradually tightened many researchers and industrialists are seeking technologies to cope with them. In order to meet the stringent regulations, research is being actively conducted worldwide to reduce NO$_x$-causing pollution. However, different countries tend to have different research trends because of their regional and industrial environments. In this paper, the results of recent catalyst studies on NO$_x$ removal by selective catalytic reduction are reviewed with the sources and regulations applied according to the national characteristics of South Korea. Specifically, we emphasized the three major NO$_x$ emissions sources in South Korea such as plant, automobile, and ship industries and the catalyst technologies used.

**Keywords:** environmental regulation; emission gas; nitrogen oxide; selective catalytic reduction; NO$_x$ removal catalyst

---

## 1. Introduction

Climate change as a result of the promotion of industrialization and urbanization through economic growth around the world has increased awareness of the atmosphere. As a result, regulations are being tightened to release air pollutants to protect the air environment and the world faces mandatory challenges to reduce air pollutants [1–6].

Figure 1 shows the results of the measurement of particulate matter (PM 2.5) exposure concentration in Organization for Economic Cooperation and Development (OECD) countries in 2017. The average PM 2.5 concentration from Figure 1 was about 12.5 μg/m$^3$, which actually exceeds the average value of 34%, and South Korea recorded the highest concentration of PM 2.5 [2].

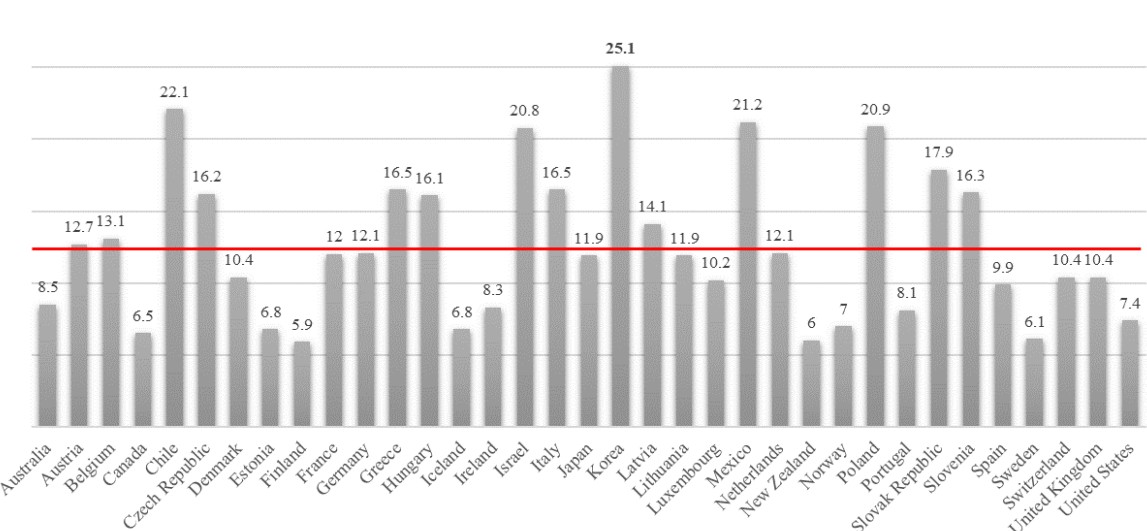

**Figure 1.** Concentration of PM 2.5 as per the Organization for Economic Cooperation and Development (OECD) countries in 2017 (Units: μg/m³).

The atmospheric condition of South Korea is threatened by the increase of high concentration of fine dust. In particular, in March 2019, Seoul's monthly average PM 2.5 concentration reached its highest level since the observation, and an emergency reduction measure for high concentrations of fine dust was issued for seven consecutive days. With this issue, the human body is not only exposed to fine dust, but also crops and ecosystems and so on. A survey on the severity of fine dust among the people found that about 30% face very serious problems, and more than 60% are serious [3]. The public's anxiety also seems to increase because fine dust, an air pollutant, is the most easily and frequently encountered part of life. In response to this anxiety, the government's response is also strengthening in South Korea.

As a result of this, the world is working to reduce the substances that cause air pollution. BBC research predicts that the management market to reduce air pollution will grow by 2019, with DeNO$_x$ systems the highest growth rate of 6.3% (Table 1). This aims to prevent the generation of additional particulate matter through the secondary chemical reaction of precursor materials (SO$_x$, NO$_x$, etc.) as well as the direct release of particulate matter.

**Table 1.** Recent history on global air pollution control equipment market by product type ($ millions).

| Type | 2012 | 2013 | 2014 | 2019 | CAGR *% 2014–2019 |
|---|---|---|---|---|---|
| Flue gas desulfurization | 14,882.1 | 15,523.1 | 16,850.2 | 21,573.0 | 5.1 |
| Electrostatic precipitators | 11,781.1 | 12,152.9 | 12,512.3 | 15,893.1 | 4.9 |
| DeNO$_x$ systems | 10,482.0 | 11,089.9 | 12,538.3 | 17,044.0 | 6.3 |
| Fabric filters | 10,139.4 | 10,478.5 | 11,685.4 | 15,101.6 | 5.3 |
| Scrubbers | 4284.8 | 4324.1 | 4245.7 | 5230.3 | 4.3 |
| Others | 2877.6 | 2985.1 | 3106.3 | 3594.7 | 3.0 |
| Total | 54,447.0 | 56,553.6 | 60,938.2 | 78,436.7 | 5.2 |

* CAGR: Component Annual Growth Rate.

Nitrogen oxide (NO$_x$) is one of the secondary sources of fine dust (components: 59.3% of sulphates, nitrates, etc., carbon and soot 16.8%, minerals 6.3%, and other substances 18.6%), which causes chemical reactions in the atmosphere. The air pollution process due to NO$_x$ is summarized in Figure 2. After

reaction with water, $NO_x$ forms nitric acid in the atmosphere, causing acid rain, and in addition, photochemical smog, tropospheric ozone, and ozone layer destruction [1,5,6].

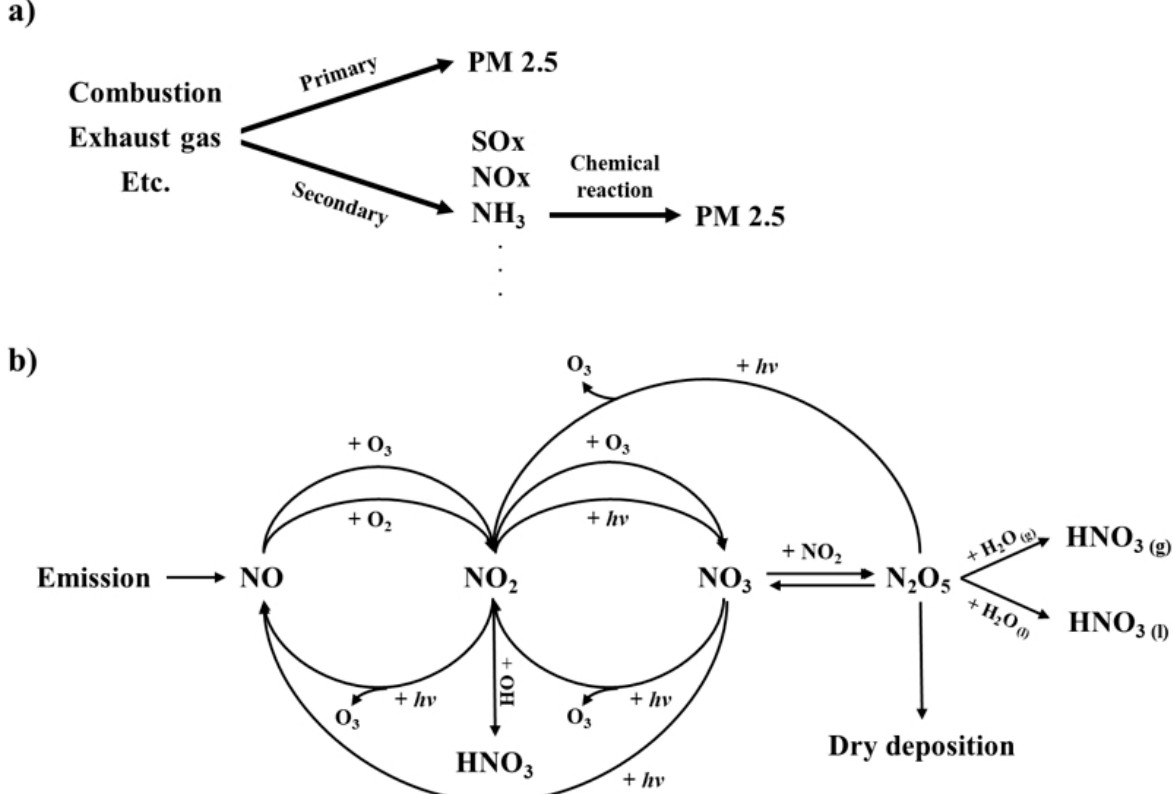

**Figure 2.** (**a**) PM 2.5 formation (**b**) atmospheric $NO_x$ reactions [1].

The causes of $NO_x$ emissions vary, so it is important to develop technologies to reduce $NO_x$, taking into account the local and industrial characteristics of each country. In South Korea, fine dust concentration mainly result from the influence of neighboring countries [7]. However, in order to reduce air pollutants emitted according to the industrial characteristics of South Korea, it is necessary to solve the problem through technology development.

In this paper, we summarized the selective catalytic reduction (SCR) reaction that is commonly used and commercialized among many $DeNO_x$ technologies. In order to examine trends in $NO_x$ reduction technology in consideration of the regional and industrial characteristics of South Korea, the changes in emission and environmental regulations according to emission sources in South Korea are examined. Moreover, the SCR catalytic technologies for $NO_x$ removal in South Korea and the future research directions are discussed.

## 2. Selective Catalytic Reduction (SCR) for Nitrogen Oxide ($NO_x$) Removal

There are several technologies under investigation to remove $NO_x$, but the most standard removal of $NO_x$ is the SCR process [8]. The SCR process is one of catalytic reactions, where single metal oxide, mixed metal oxides and supported metal oxides, and pure or metal-exchanged zeolites catalysts are generally used to convert nitrogen oxides ($NO_x$) into nitrogen molecule ($N_2$) and water ($H_2O$), which are harmless not only to human being but also the environment, by using suitable reducing agents such as ammonia, urea, hydrocarbon and hydrogen [9]. Although these reducing agents have their own advantages and disadvantages, they induce a strong decrease in the Gibbs free energy values. Likewise, the free energy values can be further reduced by the introduction of oxygen [10]. In order to protect the injected reducing agent, the SCR system is generally kept after a diesel oxidation catalytic bed. The reducing agents used for SCR process are summarized below with their mechanism.

### 2.1. Ammonia (NH$_3$-SCR Process)

The most common SCR process is the use of ammonia (NH$_3$) as the reducing agent. NH$_3$-SCR process is already commercialized around the world including South Korea and has many advantages such as flexible operating temperature, excellent redox properties and high thermodynamic stability over other reducing agents [11]. NH$_3$-SCR reaction may varies depending on the content of NO and NO$_2$ in the NO$_x$ of the exhaust gas and the main chemical reactions in NH$_3$-SCR process are given below [12]:

$$4NO + 4NH_3 + O_2 \rightarrow 4N_2 + 6H_2O \tag{1}$$

$$2NO_2 + 4NH_3 + O_2 \rightarrow 3N_2 + 6H_2O \tag{2}$$

$$6NO + 4NH_3 \rightarrow 5N_2 + 6H_2O \tag{3}$$

$$6NO_2 + 8NH_3 \rightarrow 7N_2 + 12H_2O \tag{4}$$

$$NO + NO_2 + 2NH_3 \rightarrow 2N_2 + 3H_2O \tag{5}$$

Equations (1)–(5) are reactions of NH$_3$ and NO and NO$_2$ in the presence and absence of O$_2$, respectively, that result in decomposition of NO$_x$. The standard reaction is Equation (1), which occurs when NO$_x$ consists only of NO. If there is only NO$_2$ in the exhaust gas, NH$_3$ reacts as in Equation (4), which is generally a slow reaction. In the case of Equation (5), NO$_x$ substance is a fast reaction that occurs when NO and NO$_2$ are mixed in equimolar.

Many researchers suggested that metal oxides-based catalysts having both Lewis and Brönsted acid sites are good for the NH$_3$-SCR process [13]. NH$_3$ can be adsorbed on Lewis and/or Brönsted acid sites of the metal oxides and intermediates like adsorbed NH$_3$ or NH$_4^+$ formed on Brönsted acid sites which were reacted with NO through Langmuir–Hinshelwood (L-H) mechanism producing N$_2$ and H$_2$O. The physically adsorbed NO is oxidized by metal ions to form nitrate as shown in the following reaction pathway (Figure 3) [14,15].

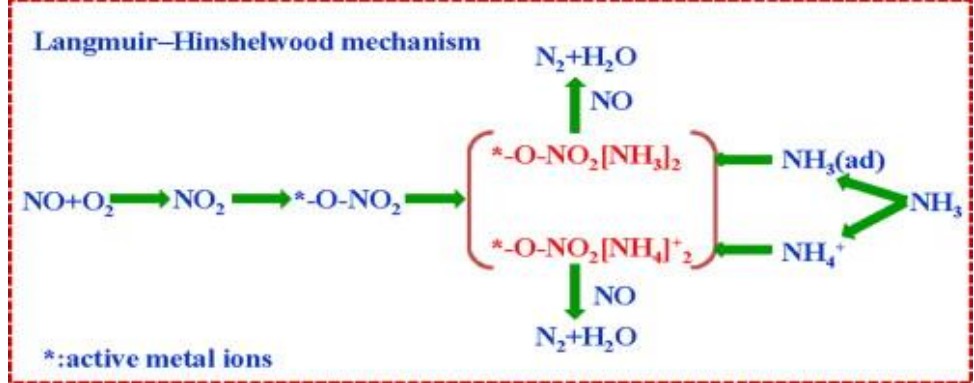

**Figure 3.** General reaction pathway for ammonia selective catalytic reduction (NH$_3$-SCR) on metal oxide catalysts. (Reprinted with permission form [15]. Copyright (2011) Elsevier).

The NH$_3$-SCR reaction mechanism (Figure 4) was proposed by Liu et al. for the FeTiO$_x$ catalyst using different analytical techniques such as in-situ DRIFTS, transient response experiments and temperature programmed desorption of NH$_3$ [16]. In more detail, Brönsted acid sites were higher than Lewis acid sites at 200 °C, whereas the former sites transformed into Lewis acid sites at >200 °C through dehydroxylation on FeTiO$_x$. L–H mechanism is proposed at 200 °C (Figure 4A). NO oxidized into nitrate species on Fe$^{3+}$ which was reacted with adjacent adsorbed NH$_3$ species on Ti$^{4+}$–OH Brönsted acid sites to intermediate species (like ammonium nitrate), followed by a subsequent reaction with NO to produce N$_2$ and H$_2$O. At high temperature, a typical Eley–Rideal (E–R) mechanism is proposed for NH$_3$-SCR of NO$_x$ (Figure 4B). In this mechanism, Fe$^{3+}$ firstly reduced to Fe$^{2+}$, which then re-oxidized to Fe$^{3+}$ by O$_2$, to complete a redox cycle. In parallel, adsorbed NH$_3$ species activated into −NH$_2$ species

by $Fe^{3+}$ via dehydrogenation, which react with NO to form an intermediate ($NH_2NO$), followed by a subsequent conversion into $N_2$ and $H_2O$.

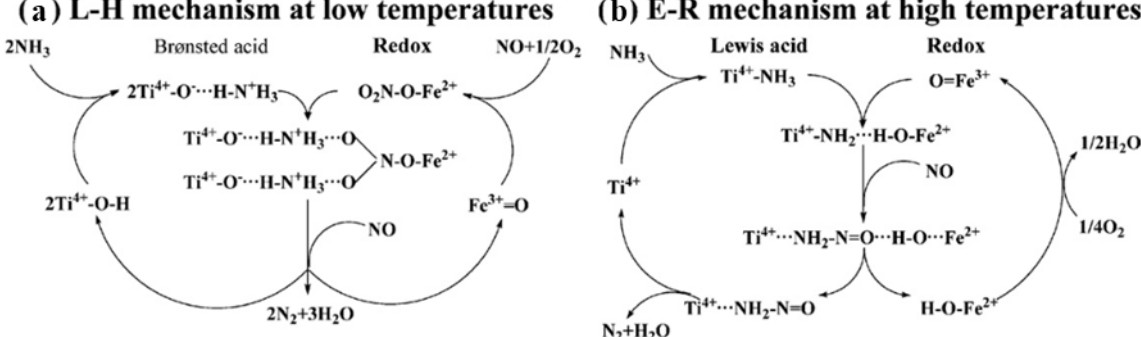

**Figure 4.** Representative $NH_3$-SCR mechanisms over FeTiOx catalyst at low (**a**) and high (**b**) temperatures. (Reprinted with permission form [16]. Copyright (2011) Elsevier).

Beside metal oxide catalysts, protonic and metal incorporated zeolites were also used for the $NH_3$-SCR process whose reaction mechanism is explained well in a previous review paper [13]. For a representative reaction mechanism on zeolite-based catalysts, $NH_3$-SCR reaction for $NO/NO_2$ over a commercial Fe–ZSM-5 catalyst was shown in Figure 5. Results were demonstrated that the intermediates of $NH_4NO_2$ or $NH_4NO_3$ species, in which $NH_4NO_2$ decomposes to $N_2$ and $H_2O$. On the other hand, $NH_4NO_3$ can directly decomposed to generate $N_2O$ and $H_2O$ [17].

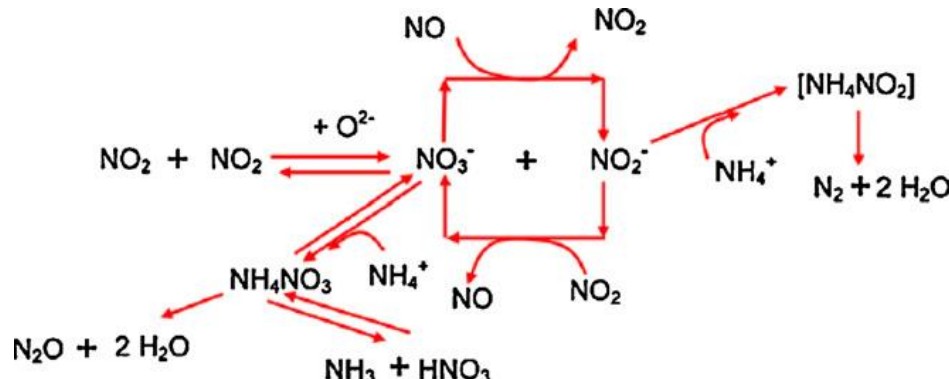

**Figure 5.** Proposed reaction scheme for NO and $NO_2$ $NH_3$-SCR over commercial Fe–ZSM-5. (Reprinted with permission form [17]. Copyright (2008) Elsevier).

Although this $NH_3$-SCR process is a well-established technology in the various parts of South Korea (for instance Pyeongtaek, Incheon and Ulsan) [18], the use of an expensive reductant, and transportation and storage of $NH_3$, besides some toxic unwanted products, are problems related to the process [8,19].

## 2.2. Urea (Urea-SCR Process)

To avoid handling hazardous and corrosive $NH_3$, urea can be used as a reducing agent that is easy to handle as aqueous solutions (Iskan City of South Korea used 40% aqueous urea-based SCR technology [18]). Urea, a large-scale commodity chemical, was considered as an $NH_3$ precursor for SCR technology. Generally, urea solution decomposes at elevated temperature to $NH_3$ in the hot exhaust gas via two-step reactions as shown in the following Figure 6:

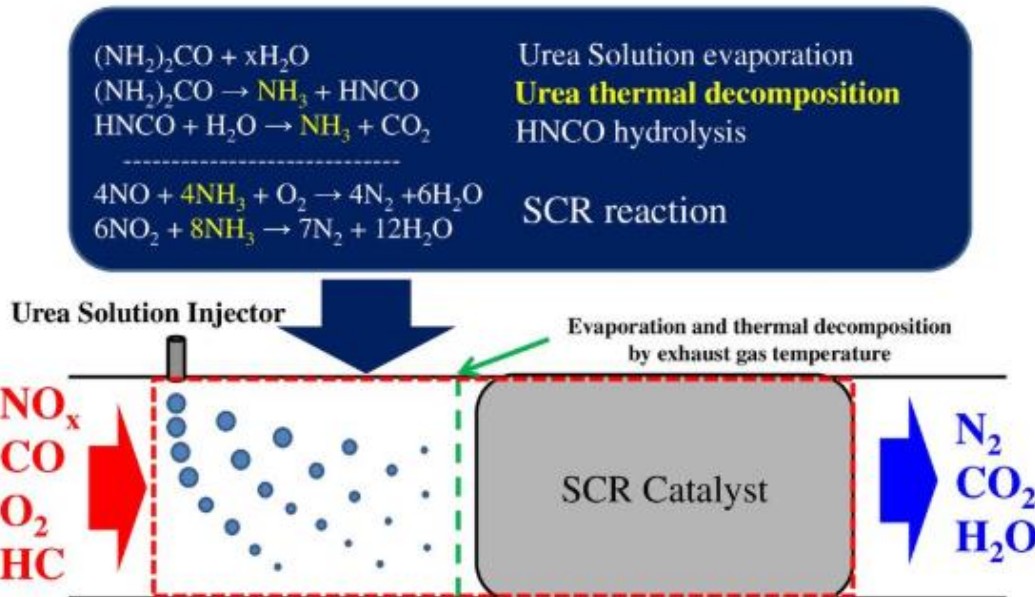

**Figure 6.** General reaction pathway for urea-SCR process [20].

Also, as shown in Figure 7, the concept adopted by Kwangsung Co. Ltd. (Gimhae, Korea) for marine/offshore using urea-SCR system [21].

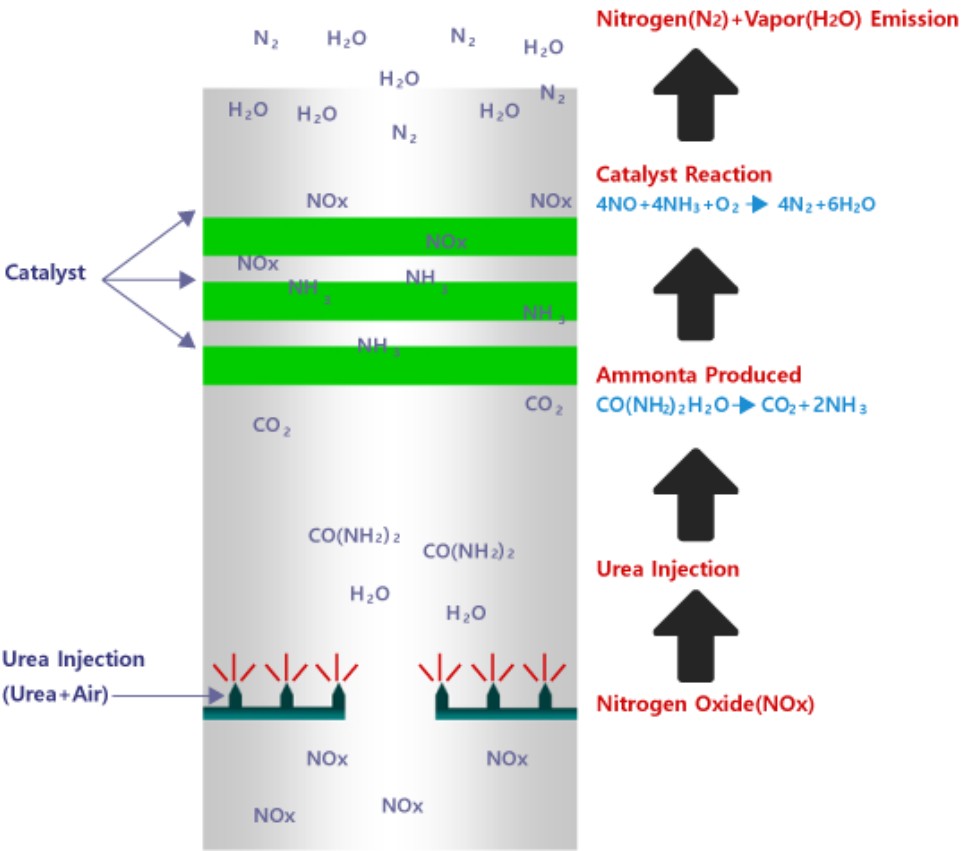

**Figure 7.** Conceptual design of Kwangsung Co. Ltd. for marine/offshore using urea-SCR process [21].

Figure 8 shows a urea decomposition reaction to ammonia and isocyanic acid (HNCO), which usually occurs under thermal reaction even without any catalysts [22], along with two byproducts namely biuret and cyanuric acid at moderate temperature of 300 °C. HNCO is a stable intermediate

species in the gas phase but hydrolyzes to $NH_3$ and $CO_2$ on the $TiO_2/ZrO_2$-based SCR catalyst. Also, smaller amounts of undesired stable products, for instance ammelide, ammeline and melamine were formed by substitution of the OH groups of cyanuric acid with one, two and three $NH_3$ groups respectively [23].

**Figure 8.** Reaction scheme for urea decomposition, including the two byproducts biuret and cyanuric acid [22].

Different catalytic activities were observed with and without water on urea hydrolysis and thermolysis [24,25]. The order of catalyst activities according to the reaction is:

Urea thermolysis: $TiO_2$ > H-ZSM-5 ≈ $Al_2O_3$ > $ZrO_2$ > $SiO_2$

Urea hydrolysis: $ZrO_2$ > $TiO_2$ > $Al_2O_3$ > H-ZSM-5 > $SiO_2$

Although urea is non-toxic and cheaper than $NH_3$, aqueous solution of urea is more difficult to dose and to mix with the exhaust gas than $NH_3$ which make urea decompose incompletely and deactivate the SCR catalyst [22].

## 2.3. Hydrocarbon (HC-SCR Process)

Hydrocarbons added to or contained in emissions can also be used as an alternative reducing agents [26]. Hydrocarbons such as methane, propane and propene appear to have excellent catalytic activity, and in particular, the selectivity of nitrogen is excellent [9,10]. The main advantage is the use of a hydrocarbon gas or its mixture very similar to that found in exhausts besides low cost and feasibility. Generally, three major reaction steps are involved during the hydrocarbon-SCR (HC-SCR) process [27]. As shown below, (a) NO decompose into $N_2$ (Equation (6)) and then hydrocarbons attack by the surface oxygen, or hydrocarbon oxidation to adjust the oxidation state of metal ion catalyst (Equation (7)), (b) the hydrocarbon intermediate (HC*) formed is able to reduce $NO_x$ selectively (Equations (8) and (9)), and (c) $NO_2$, formed from NO and $O_2$, reduced preferentially by hydrocarbon species to form $N_2$ (Equations (10) and (11)).

$$(a) \quad 2NO \rightarrow N_2 + 2O \text{ (ads)} \tag{6}$$

$$HC + O \text{ (ads)} \rightarrow CO_2 + H_2O \tag{7}$$

$$(b) \quad HC + O_2 \text{ (or } NO_x) \rightarrow HC^* + CO_2 + H_2O \tag{8}$$

$$HC^* + NO_x \rightarrow N_2 + CO_2 + H_2O \tag{9}$$

$$(c) \quad NO + 0.5O_2 \rightarrow NO_2 \tag{10}$$

$$HC + NO_2 \rightarrow N_2 + CO_2 + H_2O \tag{11}$$

Zeolites containing a Modernite Framework Inverted (MFI) structure seems to be good for the HC-SCR process, although few studies on Ferrierite structure was also reported [19]. For instance, Mn-ZSM-5 and Fe-ZSM-5 were used to remove $NO_x$ with methane and propane as the reducing agents, respectively. Lobree et al. proposed a possible mechanism on Fe-ZSM-5 for the reduction of NO by propane ($C_3H_8$) in the presence of $O_2$ (Figure 9) [28]. $Fe^{2+}$ (actually $Fe^{2+}(OH^-)$) reacts with $O_2$ to form $Fe^{3+}(O_2^-)$. NO is oxidized to $NO_2$ which is then adsorbed on the $Fe^{3+}$ catalysts surface, followed by reaction with propane. Intermediates of CN and NCO were formed from $C_xH_yNO$, of which CN react faster with $NO_2$ to form $N_2$ and carbon-containing combustion products via the oxidation or the hydrolysis reaction [9].

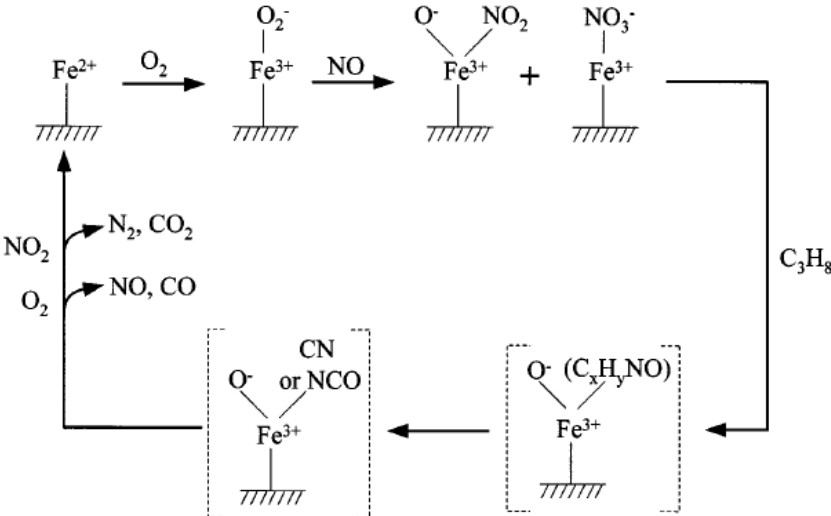

**Figure 9.** Proposed mechanism for the reduction of NO by propane ($C_3H_8$) over Fe-ZSM-5. (Reprinted with permission form [28]. Copyright (1999) Elsevier).

Some studies on alumina-based catalysts (Cu–$Al_2O_3$ and Ag–$Al_2O_3$) were also reported for the removal of $NO_x$ [29]. Nitrates obtained from $NO_x$ adsorbed on the Cu–$Al_2O_3$. The acetate species may form via the partial oxidation of propene ($C_3H_6$) with $O_2$ and possibly with nitrates. The acetate formed acts as a surface reductant that reduces nitrates to $N_2$ via the well-known possible CN and NCO intermediates (Figure 10) [30]. The different catalysts used and their deactivation phenomena for HC-SCR process can be found in a recent review article [10].

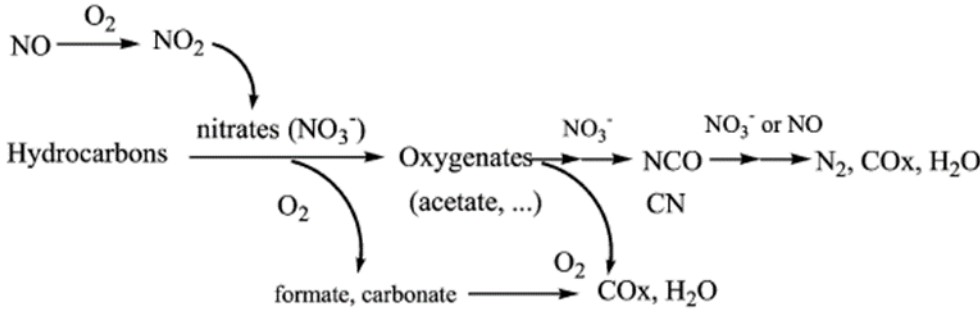

**Figure 10.** Proposed mechanism of $C_3H_6$-SCR over Cu-$Al_2O_3$ catalysts [30].

### 2.4. Hydrogen (H$_2$-SCR Process)

Compared with the NH$_3$-, urea- and HC-SCR processes, H$_2$-SCR has attracted more attention for NO$_x$ removal. Hydrogen (H$_2$), a clean environmental fuel, generates only water without inducing secondary pollutants such as CO$_2$, after its combustion with air [31]. Hydrogen can be produced from the water–gas shift reaction (CO + H$_2$O → CO$_2$ + H$_2$) in the realistic exhaust gas [32]. During the H$_2$-SCR process, the main reactions that can occur are as follows [31]:

$$2NO + 4H_2 + O_2 \rightarrow N_2 + 4H_2O \tag{12}$$

$$2NO + 3H_2 + O_2 \rightarrow N_2O + 3H_2O \tag{13}$$

$$2NO + H_2 \rightarrow N_2O + H_2O \tag{14}$$

$$NO + 5/2H_2 \rightarrow NH_3 + H_2O \tag{15}$$

$$NO + 1/2O_2 \rightarrow NO_2 \tag{16}$$

$$H_2 + 1/2O_2 \rightarrow H_2O \tag{17}$$

Equation (12) is the most desired reaction as NO reacts with H$_2$ to produce N$_2$ and H$_2$O. However, as per Equations (13) and (14), N$_2$O produced as a byproduct during reactions of two NO molecules with three H$_2$ molecules. NH$_3$ and/or NH$_4^+$ species were considered as an important intermediate species (Equation (15)) for the H$_2$-SCR reaction, although the presence of oxygen can inhibit its formation on selective catalysts [31,32]. Shibata et al. [33] used in-situ IR studies and found that the intensity of the NH$_4^+$ band on the Brönsted acid sites of Pt/MFI was closely related to the H$_2$-SCR catalytic activity. However, NH$_4^+$ band intensity decreased quickly when the hydrogen gas supply was cut off, attributed that NH$_4^+$ species highly active with NO and O$_2$. The NH$_4^+$ species likely to be formed between the reaction of adsorbed N, which was obtained from the dissociation of NO, and the active hydrogen on the Pt sites. The complete reaction scheme of H$_2$-SCR over Pt/MFI is shown in Figure 11.

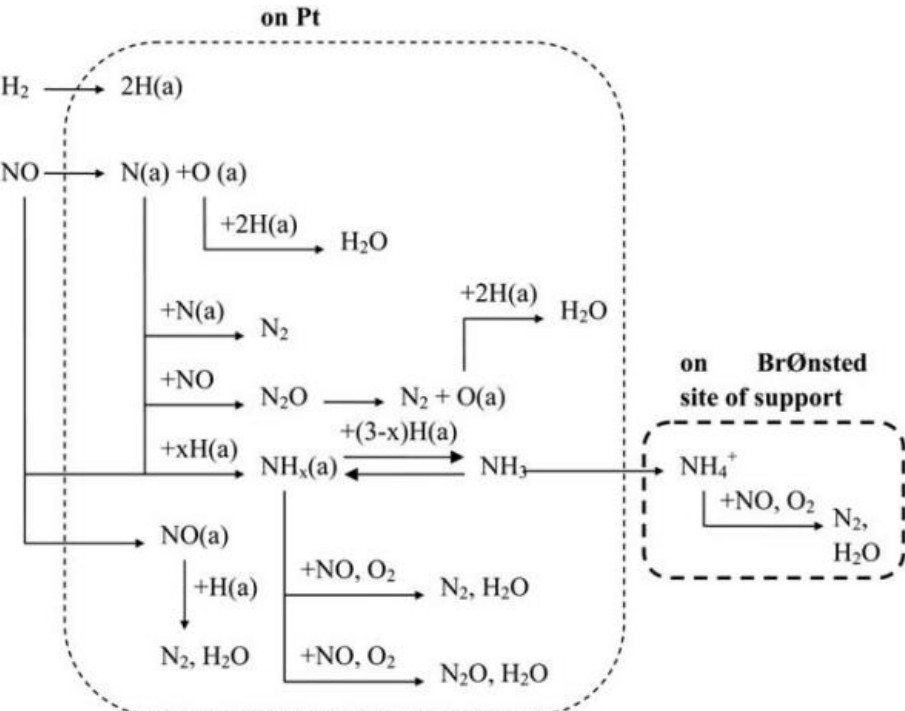

**Figure 11.** Proposed reaction scheme for H$_2$-SCR over Pt/Modernite Framework Inverted (MFI) catalyst [33].

Pt and Pd metals show a high activity for $H_2$-SCR at low temperature. The nature of supports and their intrinsic acidity greatly affect the activity of $NO_x$ reduction and selectivity to $N_2/N_2O$ [34]. The details of the active metals, and their supports and degree of deactivation rate for $H_2$-SCR were already covered in past review papers [31,32,34]. As far as we know, no South Korean industries have utilized both HC- and $H_2$-SCR processes until now. However, with strict regulation by the government, we are anticipating some initiation from heavy industries of using those processes which will reduce unwanted pollutants such as $CO_2$, a key greenhouse gas.

## 3. Recent SCR Technology Status for $NO_x$ Removal in South Korea

### 3.1. $NO_x$ Emission Sources

South Korea's Clean Air Policy Support System (CAPSS) source classification system for nitrogen oxides ($NO_x$) has been adopted based on the European CORINAIR Source Classification System (SNAP 97). The classification system has been changed to 13 major categories according to the $NO_x$ emission conditions in South Korea since 2007, and is summarized in Table 2 along with the source classification codes (SCC) and details for each item. Figure 12 shows the $NO_x$ emissions of South Korea from 2010 to 2016 according to the CAPSS source classification system [35]. As a result, total $NO_x$ emissions are estimated to increase by 31,183 tons per year from 1,061,210 ton/yr (2010) to 1,248,309 tons/yr (2016). According to the type of source, the results in 2016 show that the ratio of mobile pollutant sources (on-roads (07) and non-roads (08)) was the highest, accounting for 61% (89,000 tons), the largest increase in the last six years, which led to the increases of the total $NO_x$ emissions. $NO_x$ emitted through the combustion process (energy industry (01), non-industry (02), manufacturing industry (03)), which account for the second highest $NO_x$ emissions with about 33% of the total emissions, maintaining more than 400,000 tons annually. Other production processes (04), waste treatment (09) and cotton pollutants (11) produced approximately 78,000 tons of $NO_x$, and no $NO_x$ emissions for SCC (05), (06), (10), and (12) were calculated. The results of $NO_x$ emissions discussed above are based on air pollutant classification, and when the $NO_x$ emissions are classified by different regions of South Korea, the order of the sources that emit the $NO_x$ may partially change. Therefore, Lim et al. [36] pointed out that there is a need for an efficient air-quality management policy, not a policy focused only on urban areas, because of regional differences in emission characteristics.

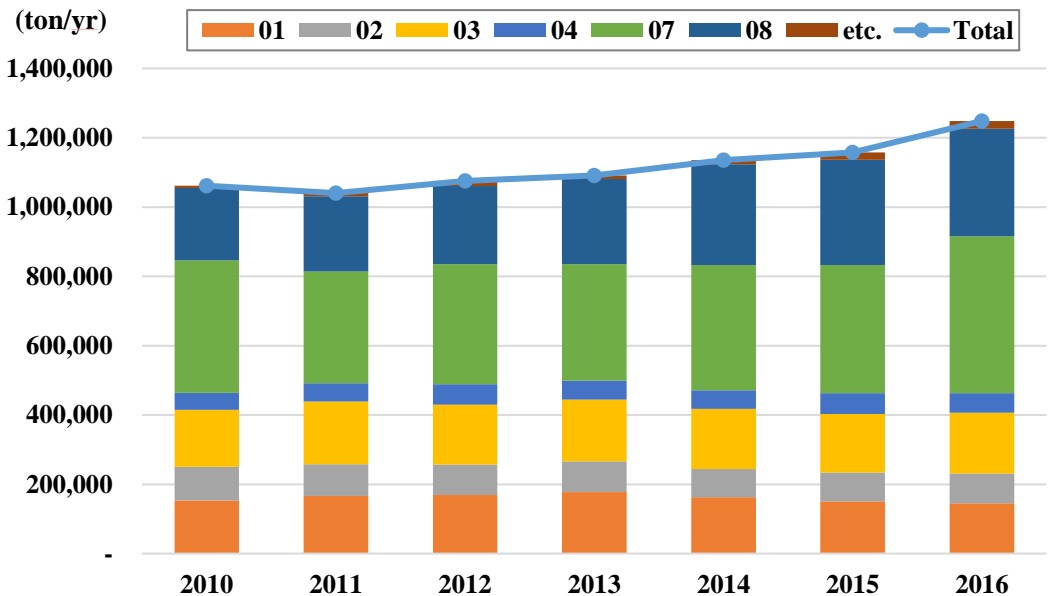

**Figure 12.** $NO_x$ emission sources in South Korea from 2010 to 2016 [35].

**Table 2.** Categories of $NO_x$ emission sources in South Korea [35].

| SCC | Major Categories of Emission Sources | Details |
|---|---|---|
| (01) | Energy industrial combustion | Public and private power generation facilities, district heating facilities, oil-refining facilities |
| (02) | Non-industrial combustion | Commercial and public institutions, residential facilities, agriculture, livestock and fisheries facilities |
| (03) | Manufacturing combustion | Combustion facilities, furnaces, etc. |
| (04) | Production process | Petroleum products, industry, steelmaking, inorganic chemicals manufacturing, organic chemicals manufacturing, wood, pulp manufacturing, food and beverage processing, other manufacturing |
| (05) | Energy transport and storage | gasoline supply |
| (06) | Use of organic solvents | Painting facilities, washing facilities, laundry facilities, other organic solvents |
| (07) | On-road vehicles | Passenger cars, taxis, vans, buses, vans, special vehicles, motorcycles |
| (08) | Non-road vehicles | Railway, ship, aviation, agricultural machinery, construction equipment |
| (09) | Waste incineration | Incineration, other waste disposal |
| (10) | Agriculture | |
| (11) | Other sources | Other things |
| (12) | Scattering dust | dust road, construction activity |
| (13) | Bio-combustion | |

SCC: Source Classification Codes.

## 3.2. $NO_x$ Environmental Regulations

Regulations for $NO_x$ in South Korea follow the legislation amended by international organizations or the Korean Ministry of Environment. In general, fixed sources such as the chemical industry and power generation units have strong regional characteristics, so they follow emission allowances issued by environmental ministries in each country, but mobile sources (i.e., on-roads and non-roads vehicles) are not limited by the regulations of one country.

The Air Environment Conservation Act promulgated by the Ministry of Environment is followed for the fixed pollution sources. Recent revisions of regulations show a significant reduction in emission limits of $NO_x$ and $SO_x$ from 2020, as compared to 2019, as shown in Figure 13. Emission allowance standards applied to workplaces may vary depending on the nature of installed facilities. Under the current law, the $NO_x$ emission limit was 20–530 ppm, but according to the revised law from 2020, the value is strengthened about 28%. The details of DeNO$_x$ facilities and their fuel type can be found in Appendix 8 of the Enforcement Regulations of the Air Quality Conservation Act (Table 3). In addition, due to the higher $NO_x$ emission level than the actual allowance standard, the air emission control panel fixed the penalty amount of 2130 won/kg of $NO_x$ [37]. In South Korea, the regulations in plant industry (boilers and power generation) were established at 2000, though the regulations are gradually strengthened every year. For instance, the allowable concentration of $NO_x$ for power generation facilities (shown in Table 3) has been significantly reduced since 2015, which is more than twice the previous year's limit of $NO_x$ emission.

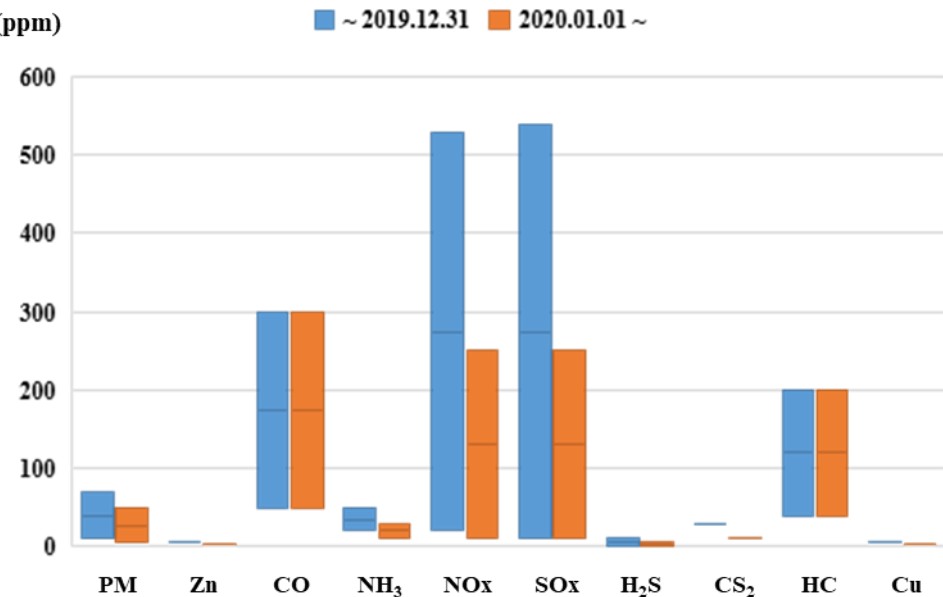

**Figure 13.** General air pollutants with emission standards in South Korea.

**Table 3.** The standard for the emission of $NO_x$ in South Korea.

| Facility | Fuel Type | Installation Period | $NO_x$ Emission Standard [ppm] |
|---|---|---|---|
| Boiler | Liquid [1] | ~2001.06.30 | 130 |
| | | 2001.07.01~2014.12.31 | 70 |
| | | 2015.01.01~ | 50 |
| | Solid | ~2007.01.31 | 120 |
| | | 2007.02.01~ | 70 |
| | Gas [1] | ~2014.12.31 | 150 |
| | | 2015.01.01~ | 40 |
| Power generation | Liquid [2] | ~2001.06.30 | 530 |
| | | 2001.07.01~2014.12.31 | 270 |
| | | 2015.01.01~ | 90 |
| | Solid [3] | ~1996.06.30 | 70 |
| | | 1996.07.01~2014.12.31 | 50 |
| | | 2015.01.01~ | 15 |
| | Gas [4] | ~2001.06.30 | 80 |
| | | 2001.07.01~2014.12.31 | 50 |
| | | 2015.01.01~ | 20 |

Source: Clean Air Conservation Act (Article 16 Appendix 8) in South Korea. [1] For the case of evaporation rate is over the 40 tons/h or calorific quantity is over the 24,760 kcal/h. [2] For case of internal combustion engine for power generation (diesel). [3] For the case of generation capacity is over the 100 MW. [4] For the case of the internal combustion engine for power generation (include gas turbine).

Mobile pollution sources, namely on-road and non-road vehicles, are the largest producers of $NO_x$ emission in South Korea. Automobiles are mainly responsible for the on-road pollution sources, whereas ships are majorly contributors to the non-road pollution sources. The emission standards for automobiles and ships in South Korea are typically in accordance with international regulations specified by the United States (US) and Europe (Figure 14a,b). Regulations for automobiles are based on the Super Ultra-Low Emission Vehicle (SULEV) standard, which is the ultra-low emission vehicle

standard for Californian gas emissions of gasoline and gas vehicles. This is the world's strongest standard, aiming to reduce 90% of the existing models. In the case of diesel cars, the European Union's standards are applied, which are in line with the recent rise in the regulatory level [37–39].

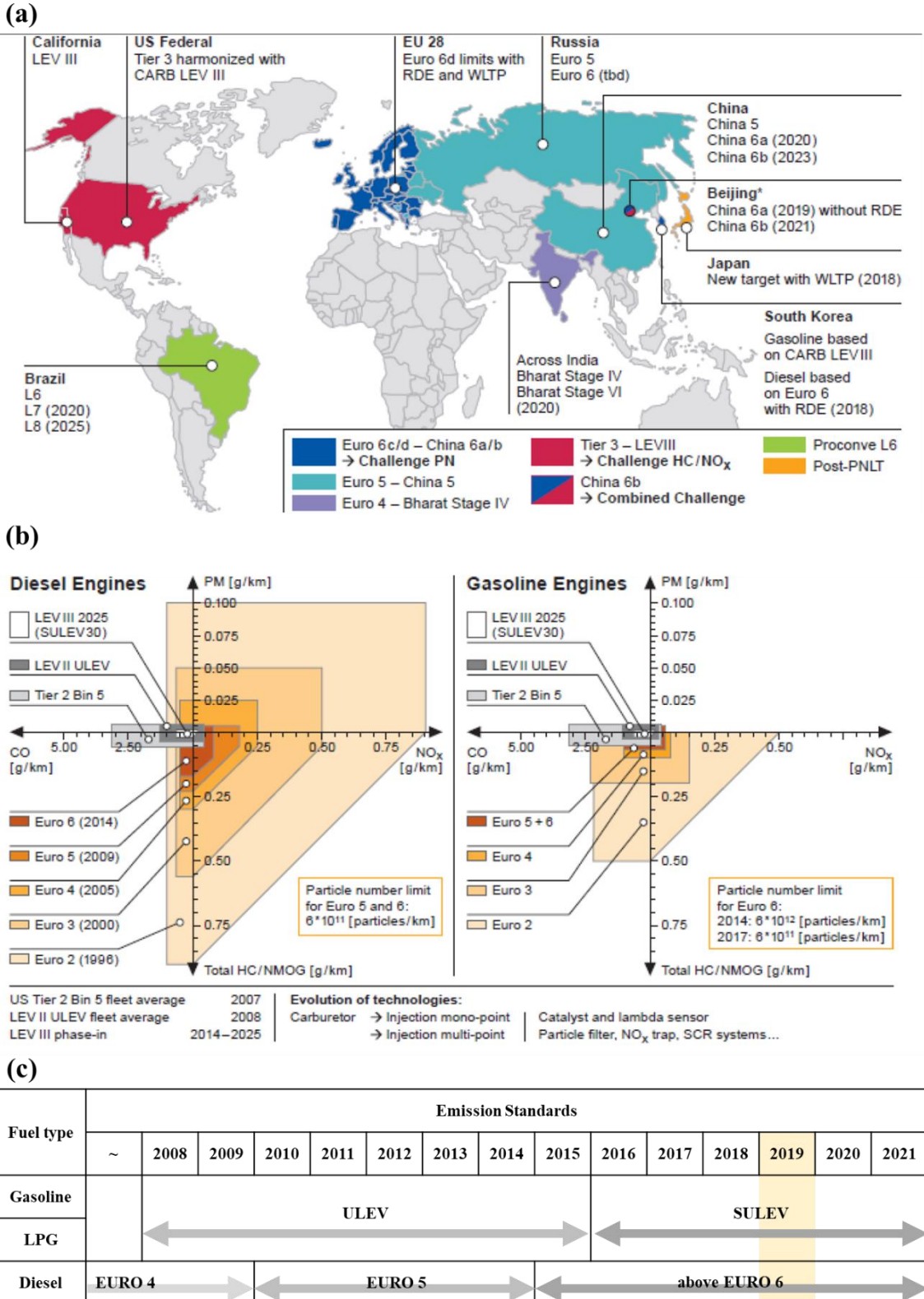

**Figure 14.** $NO_x$ Emission standard of on-road vehicles on worldwide (**a**,**b**) and South Korea (**c**) [37–39].

In 2000, the regulation standards of emission control from the United States and Europe were introduced to Korea, and they were applied almost similar to Korea since 2010. $NO_x$ emission regulations have been significantly strengthened since 2006, with the emission standards of vehicles classified as gasoline, liquefied petroleum gas (LPG) and diesel vehicles (Figure 14c). The certification test methods applied in the same New European Driving Cycle (NEDC) modes as Europe, starting with the introduction of Euro-3 in 2002 and the application of Euro-4 criteria in 2006 (Figure 15) [40].

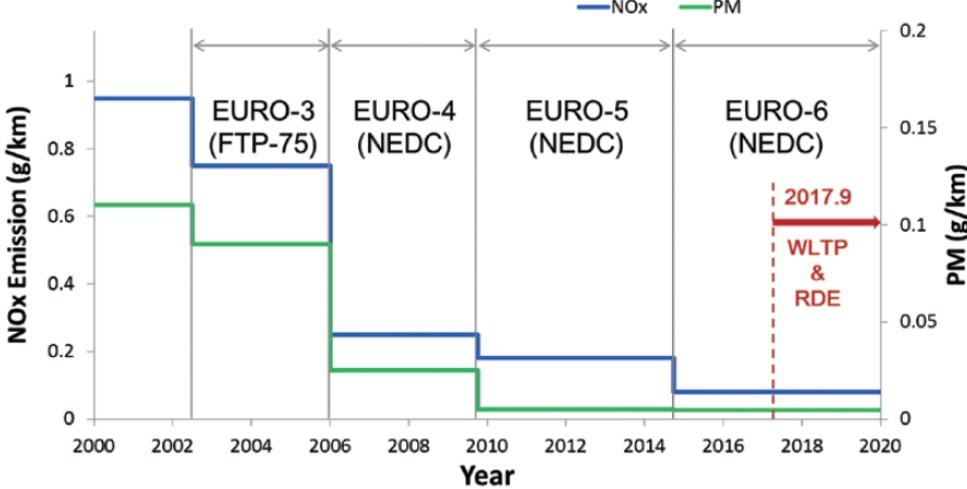

**Figure 15.** $NO_x$ emission standard for lightweight diesel vehicles in South Korea [41].

In the case of regulations applied to ships, the total tonnage of ships that enter and leave in South Korean ports was increased to 5.6% in 2016 as compared to the previous years. Of the total, the portion of oceangoing ship amounts are 89% (Figure 16). Therefore, due to the nature of foreign ships that have to pass through various countries and international waters, the regulations on the emission of $NO_x$ for domestic ships also relate to international regulations.

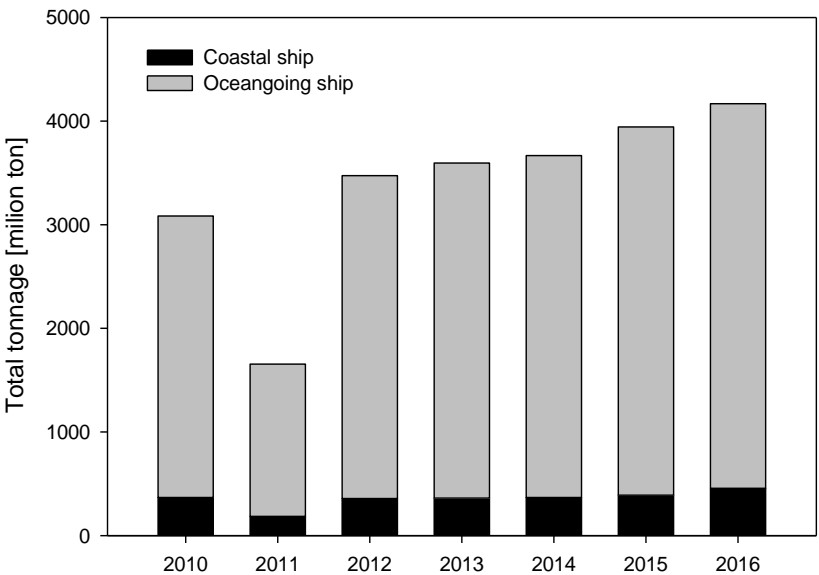

**Figure 16.** Status of ship entry and exit in South Korea [42].

As per Annex VI of the Convention on the Prevention of Marine Pollution (MARPOL), the Marine Environment Protection Committee (MEPC) of the International Maritime Organization (IMO) has applied the emission regulations of $NO_2$ from diesel engines at international ports since 1 January 2000. The regulatory standard has been applied to reduce nitrogen oxide emissions by 78%, which

may be less than 3.4 NOx g/kWh, or less than the amount emitted by the existing Tier II (Table 4 and Figure 17). In addition, according to the Annex VI of MARPOL, the North American and Caribbean waters are now designated as a nitrogen oxide control area and are considering further expansion to the North Sea and Baltic Sea [43].

**Table 4.** The status of emission regulations of $NO_x$ in the Convention on the Prevention of Marine Pollution (MARPOL).

| Period | | ~2000.01.01 | 2011.01.01~2015.12.31 | 2016.01.01~ |
|---|---|---|---|---|
| Classification | RPM (n) | Tier I | Tier II | Tier III |
| Outside ECA | $n < 130$ | 17.0 | 14.4 | |
| | $130 \le n < 2000$ | $45.0 \times n \, (-0.2)$ | $44.0 \times n \, (-0.23)$ | |
| | $2000 \le n$ | 9.8 | 7.7 | |
| ECA (North America, Puerto Rico, Virgin island) | $n < 130$ | 17.0 | 14.4 | 3.4 |
| | $130 \le n < 2000$ | $45.0 \times n \, (-0.2)$ | $44.0 \times n \, (-0.23)$ | $9.0 \times n \, (-0.2)$ |
| | $2000 \le n$ | 9.8 | 7.7 | 2.0 |

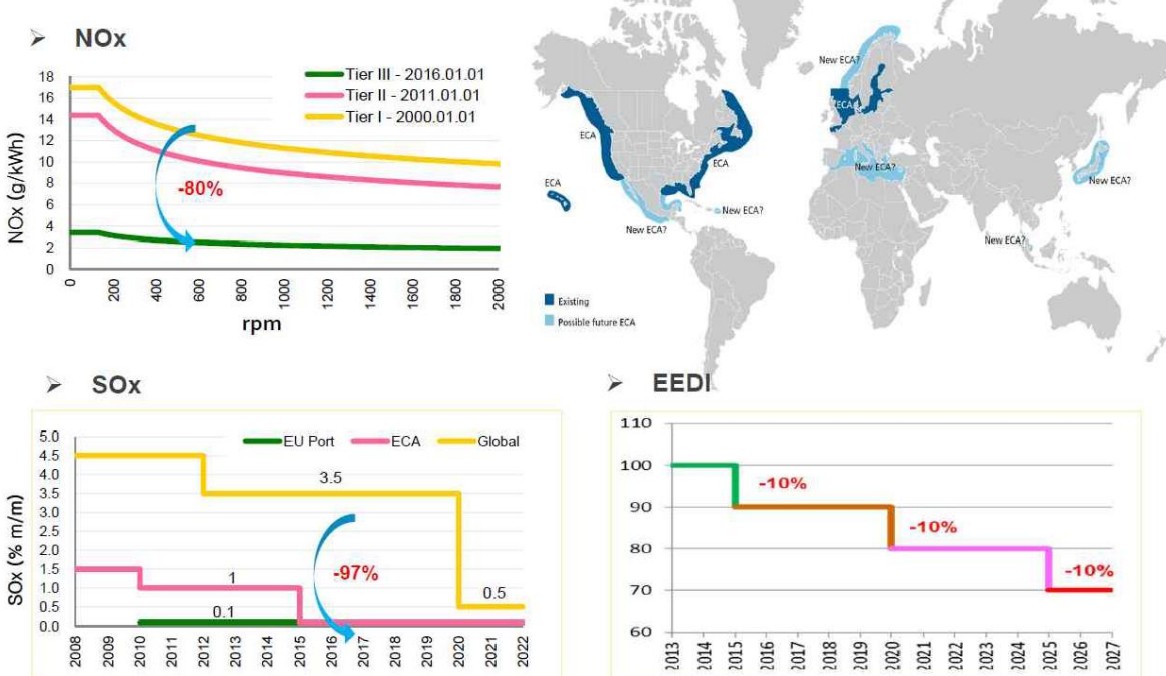

**Figure 17.** The status of $NO_x$ and $SO_x$ emission limitation [43].

In the case of the SCR system for $NO_x$ reduction, not only is the regulation different for each emission source but also the composition of exhaust gas is different. To remove $NO_x$, the catalytic technology can be classified into three broad categories: environmental catalysts (to remove $NO_x$ emitted after chemical processes), automotive catalysts (to remove $NO_x$ from automotive exhaust gases), and marine catalysts (for $NO_x$ reduction in the marine industry). Therefore, the technology trends of South Korea responding to the tightened $NO_x$ emission regulations are summarized in the following section.

### 3.3. Research Trends of SCR Catalysts for DeNO$_x$

### 3.3.1. Power Plants

In power plants, an increase of DeNO$_x$ catalyst layer is directly proportional to the efficiency of nitrogen oxide removal. The power plants of South Korea are removing nitrogen oxides in the exhaust gas using SCR method. Most SCR plant units are often designed to be equipped with four-layer catalysts, though only two layers are now filled with catalyst to satisfy the emission criteria. A four-layer SCR catalytic system also displayed a good efficiency for removal of NO$_x$. However, they suffered a pressure drop, due to additional catalytic layers, which increased not only the overall cost of the DeNO$_x$ process but also the ammonia distribution in the nozzle of ammonia injection facility [44].

According to the Korea Electric Power Exchange's statistics in 2018, the capacity of thermal power plants using gas and coal as raw materials accounts for about 69% of the total capacity (Table 5). While the number of thermal power generators using gas and coal as raw materials, accounting for 73% of the total number of main power generators. With this power industry environment, the thermal power has a higher number of operating days as compared to other sources of power generation, especially 81.53% for coal-fired power generation (Table 6) [45].

**Table 5.** Status of main power generator in South Korea [45].

| Classification | The Capacity of Facility [MW] | Number of Generator |
|---|---|---|
| Nuclear | 21,850 | 23 |
| Coal | 35,408 | 61 |
| Gas | 37,837 | 236 |
| Water | 6282 | 57 |
| Oil | 4104 | 31 |
| Total | 105,481 | 408 |

**Table 6.** Rate of operation by power generator in South Korea [45].

| Classification | Total | Nuclear | Coal | Gas | Water | Oil |
|---|---|---|---|---|---|---|
| 2014 | 70.01% | 82.39% | 90.07% | 61.53% | 20.32% | 27.81% |
| 2015 | 66.85% | 84.82% | 90.33% | 48.82% | 16.12% | 38.02% |
| 2016 | 67.80% | 81.26% | 87.54% | 49.85% | 26.86% | 48.52% |
| 2017 | 65.33% | 73.82% | 85.44% | 50.32% | 30.66% | 31.95% |
| 2018 | 63.98% | 65.31% | 81.53% | 55.16% | 37.68% | 26.39% |

The exhaust gas treatment facility of coal-fired power generation requires stable performance for a long period of time. In order to preserve long-term stability, the presence of proper catalysts in coal-fired power-generating SCR facilities is highly recommended. Therefore, long-term stability of operation is required for catalyst injection into power-generating SCR facilities. In addition, for boilers used in thermal power generation, the temperature of the boiler exhaust is higher than 500 °C which is enough to operate any catalyst. However, due to its higher temperature, the SCR catalyst may deactivate during long-term operation. Therefore, since 2000s, when SCR facilities were supplied to South Korea, various studies have been conducted for stable operation of the SCR catalyst.

The catalysts used in the SCR process are $V_2O_5$-$WO_3$ (or $MoO_3$)/$TiO_2$ have honeycomb and plate type structures (Figure 18). In particular, the $V_2O_5$-$WO_3$ (or $MoO_3$)/$TiO_2$ monolith catalyst of the honeycomb type has been consistently used as it has advantages such as high heat and mass transfer rates besides high contact area [46].

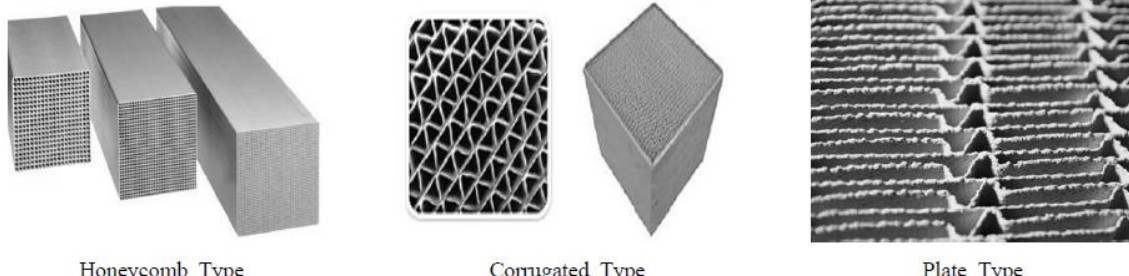

**Figure 18.** General types of SCR catalyst [47].

Looking at the amount of installation of SCR catalyst in South Korea in 2015 (Table 7), the subsidiary power plant of the Korea Electric Power Corporation (KEPCO) is approximately 21,758 m$^3$, and third-party power plant usage is estimated to be around 4786 m$^3$. The total use of SCR catalyst for domestic power plants and incinerators is roughly around 26,660 m$^3$ [47].

**Table 7.** The status of installed SCR catalyst in South Korea (2015) [47].

| Classification | Cat. Type | Total Amount of Installed SCR Catalysts [m$^3$] |
|---|---|---|
| KEPCO and Subsidiary | Honey comb | 6462 |
| | Plate | 13,125 |
| | Corrugate | 2171 |
| Other company | - | 4786 |
| Incineration plant | - | 95 |
| Total | | 26,660 |

Huang et al., according to V$_2$O$_5$ content, investigated the thermal behavior of V$_2$O$_5$/TiO$_2$ catalyst. They changed the pretreatment temperature of the catalyst from 250 to 500 °C. Through this experiment, they reported that at temperatures above 400 °C the phase transition of V$_2$O$_5$ occurred and condensation caused the decrease of the specific surface area (Figure 19) and hence of the catalytic activity [48].

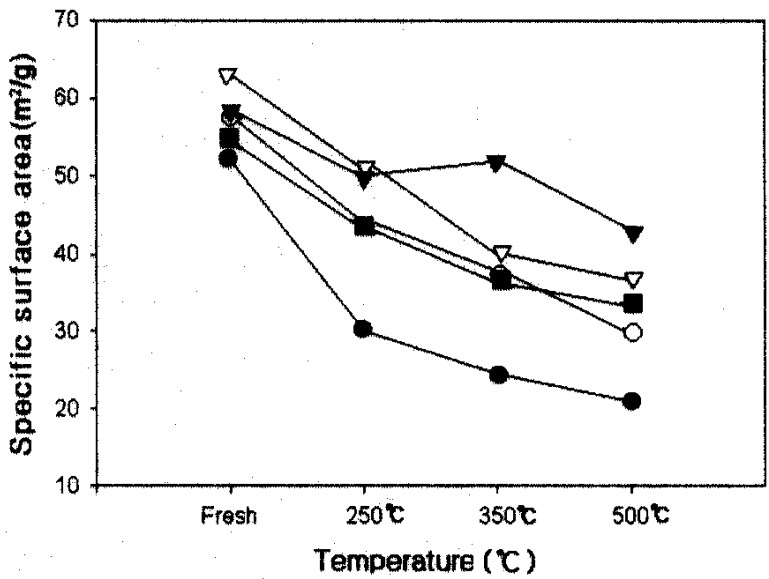

**Figure 19.** Effect of catalyst sintering temperature on specific surface area: (■) 0 wt%, (●) 1.0 wt%, (○) 3.0 wt%, (▼) 5.0 wt% and (▽) 10.0 wt% with V$_2$O$_5$ content in the catalyst [48].

Cha et al. investigated the thermal deactivation characteristics of $V_2O_5$-$WO_3$/$TiO_2$ catalyst of plate-type. They reported that when the catalyst calcined at temperatures above 600 °C, the $NO_x$ conversion rate was dramatically reduced (Figure 20). The XRD analysis results showed that the crystalline structure of $TiO_2$ changed from anatase to rutile, and the surface area and the contribution of the catalyst was significantly reduced at those temperatures due to the formation of $CaWO_4$ and $TiO_2$ crystals, which were confirmed by scanning electron microscope (SEM) images (Figure 21) [49].

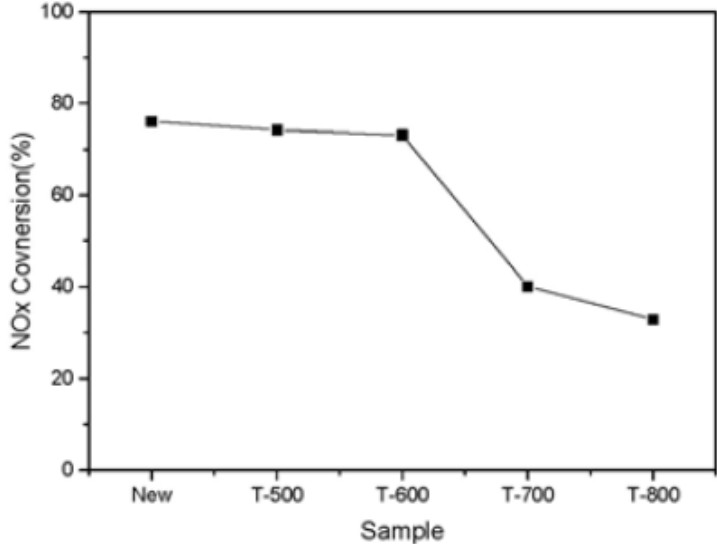

**Figure 20.** $NO_x$ conversion of plate type SCR catalyst at different calcination temperature [49].

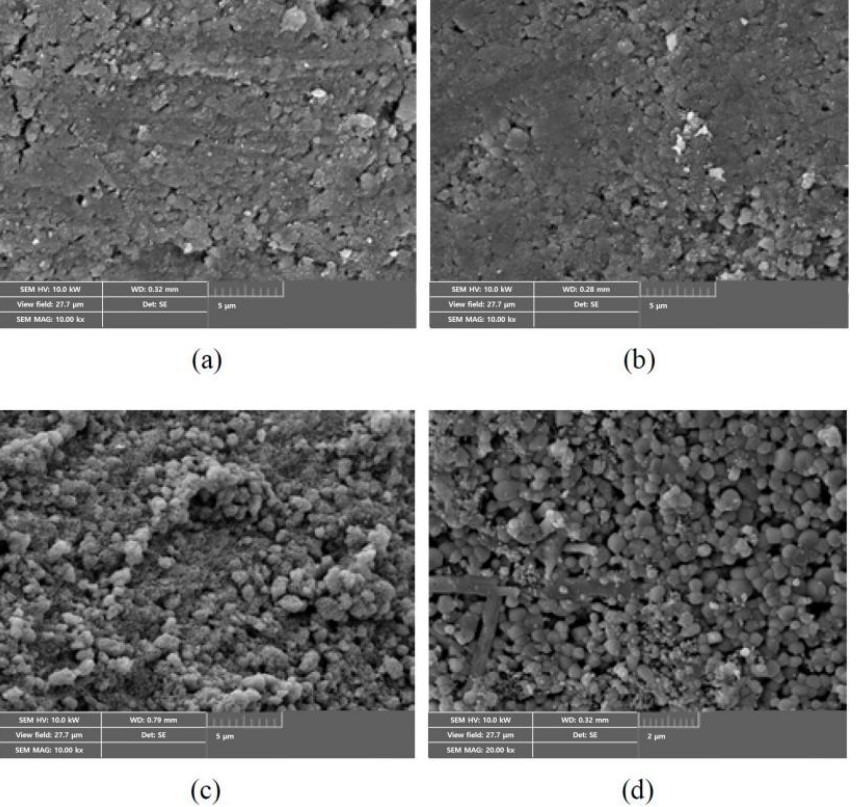

**Figure 21.** Microstructure of catalysts surface at different calcination temperature: (**a**) new, (**b**) T-600, (**c**) T-700, (**d**) T-800 [49].

Nam et al. conducted a study on the high-temperature SCR reaction using W/TiO$_2$ catalyst. They found that TiO$_2$ alone showed a conversion rate of NO$_x$ of less than 50% at temperatures above 500 °C. However, addition of W with 13 wt% into TiO$_2$ showed an increased SCR efficiency, but more than 13 wt% content of W on TiO$_2$ led to lower efficiency. W(13)/TiO$_2$(B) (where W content is 13 wt% and B = pure anatase) catalyst was pretreated at 600 °C and then loaded SCR reaction operated at 500 °C and 550 °C. The catalytic activity was stable even after more than 900 h (Figure 22) [50].

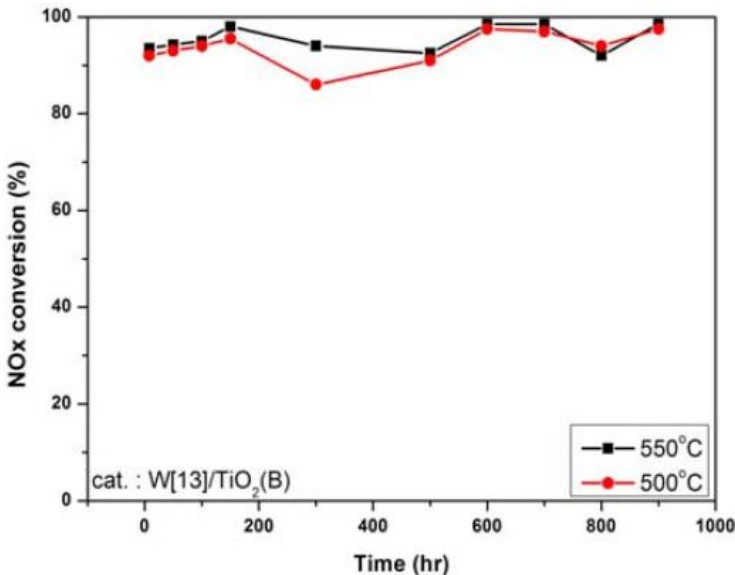

**Figure 22.** The effect of thermal aged on NO$_x$ conversion over W [13]/TiO$_2$(B) (Reaction condition: NO$_x$: 400 ppm, NH$_3$/NO$_x$: 1.0, O$_2$: 8 vol%, H$_2$O: 8 vol%, Space velocity (S.V): 60,000 h$^{-1}$) [50].

Except for liquefied natural gas (LNG), most fossil fuels contain significant amount of sulfur dioxide within the exhaust gas after combustion of these fuels. SO$_2$ may be oxidized into SO$_3$ in the presence of catalysts, which converted again into SO$_4^-$ species on the catalyst surface. The new species formed inhibits the catalytic active sites, or SO$_4^{2-}$ species may be combined with NH$_4^+$ to create salt, NH$_4$HSO$_4$. This salt also made it possible to cover the catalyst active species, and thus greater catalytic deactivation was observed in the presence of SO$_2$ [51].

Lee et al. carried out studies to enhance SCR reactivity in the presence of SO$_2$ on the V/TiO$_2$ catalyst. They showed different SCR conversion rates and SO$_2$ durability results depending on the vanadium content. The NO$_x$ conversion was increased by increasing the vanadium amount (from 0.5 to 3.0 wt%) on TiO$_2$, whereas the relative activity for SCR of NO$_x$ in the presence of SO$_2$ followed an opposite trend (Figure 23). They have also tested the relative activity for SCR after the addition of 5 wt% W into the optimized V[2]/TiO$_2$ (i.e., V[2]W[5]/TiO$_2$) catalyst which displayed an improved performance of NO$_x$ in the presence of SO$_2$ 500 ppm (Figure 24a) [52]. It was observed from SO$_2$-Temperature Programed Desorption (TPD) (Figure 24b) that the desorption peak was shifted to lower temperature on V[2]W[5]/TiO$_2$ catalyst as compared with that of W-free catalyst (V[2]/TiO$_2$).

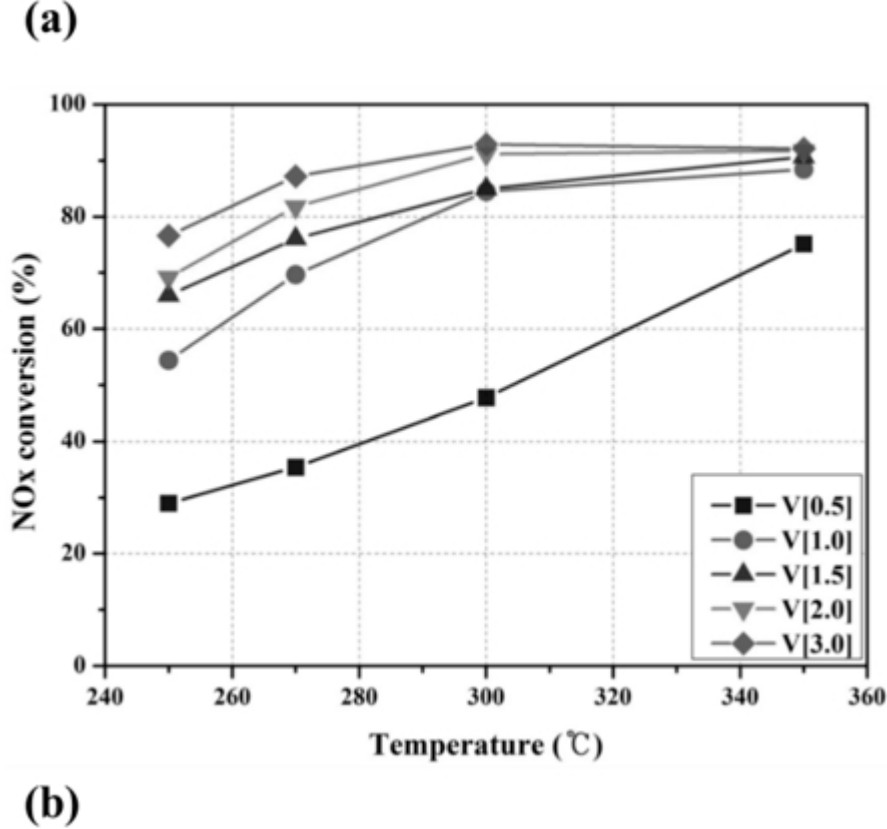

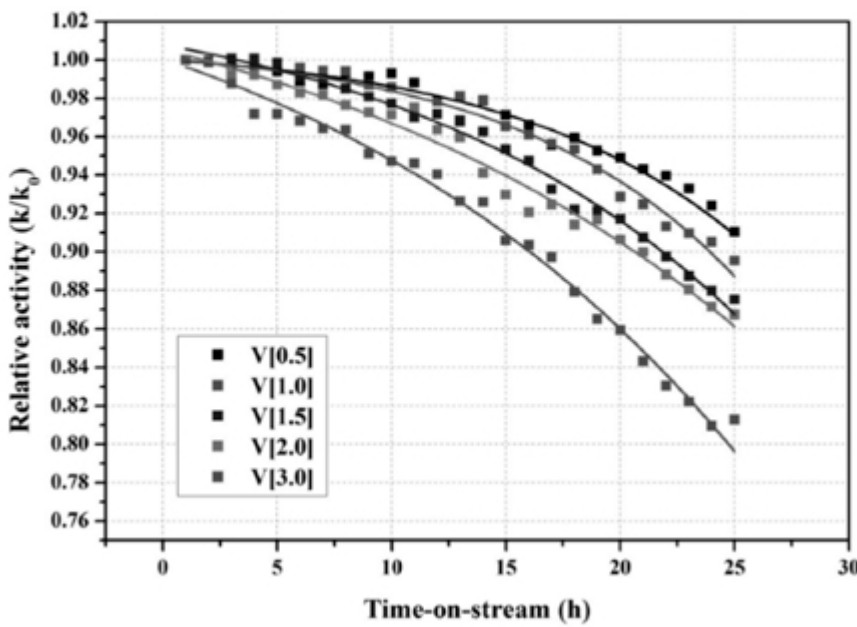

**Figure 23.** (**a**) Effect of vanadium contents of V[x]/TiO$_2$ catalysts on NO$_x$ conversion (reaction conditions: NO: 748 ppm, NO$_2$: 55 ppm, O$_2$: 3%, H$_2$O: 6%, NH$_3$/NO$_x$: 1.0, S.V: 60,000 h$^{-1}$). (**b**) Relative activity in the present of SO$_2$ for the SCR of NO by NH$_3$ with V[x]/TiO$_2$ catalysts [52].

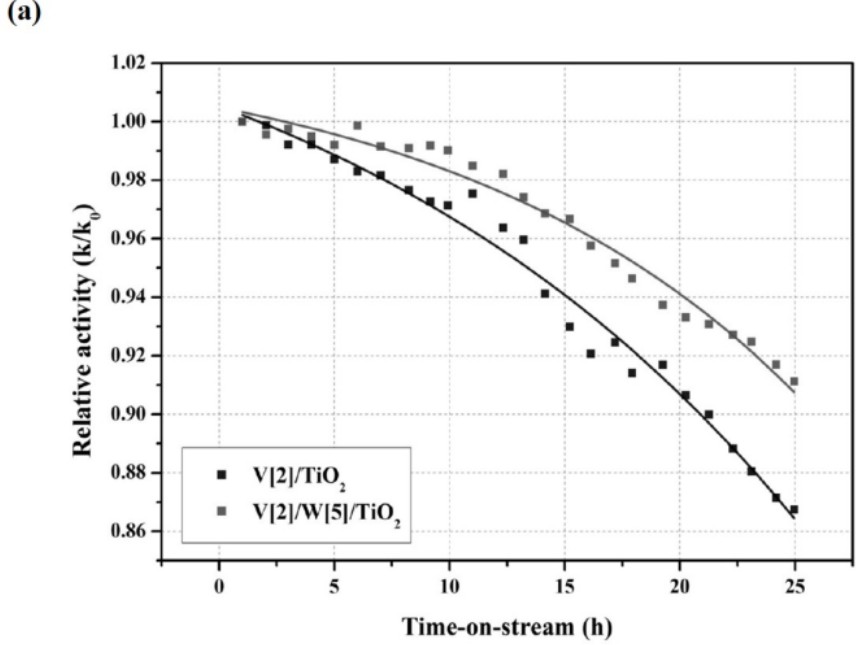

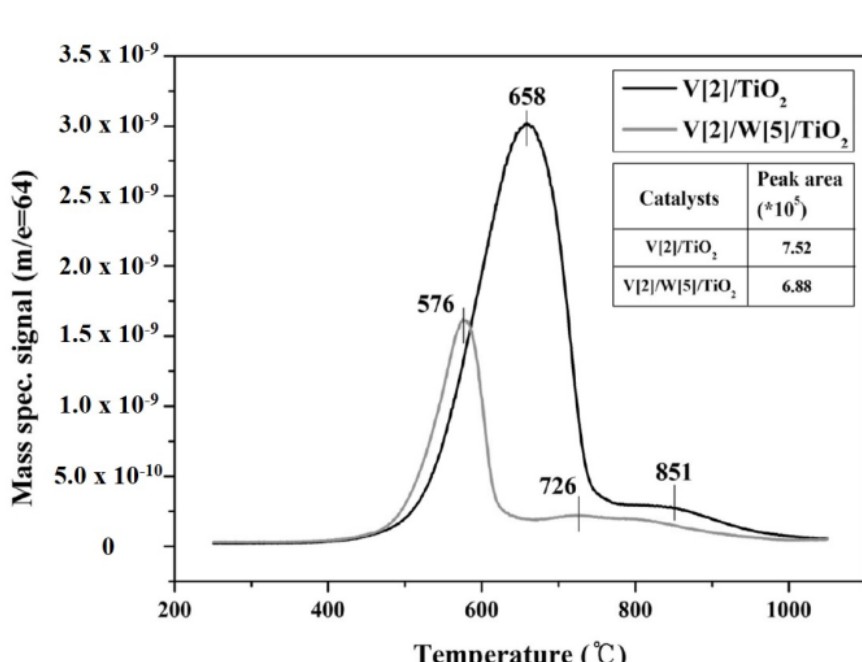

**Figure 24.** (**a**) Relative activity in the present of $SO_2$ for the SCR of NO by $NH_3$ with $V[2]/TiO_2$ and $V[2]/W[5]/TiO_2$ catalyst(reaction conditions: reaction temperature: 250 °C, NO: 748 ppm, $NO_2$: 55 ppm, $O_2$: 3%, $H_2O$: 6%, $SO_2$: 500 ppm, $NH_3/NO_x$: 1.0, S.V: 60,000 $h^{-1}$) (**b**) $SO_2$-TPD pattern of $V[2]/TiO_2$ and $V[2]/W[5]/TiO_2$ catalysts. [52].

Park et al. found that over 90% of catalytic activity was recovered when spent catalyst was washed by acidic solution in the field regeneration of SCR waste catalyst, and the concentration of acidic solution was 3~5 M (Figure 25). When the acidity of the solution exceeds the optimal point, the dissolution effects of the impurity component deposited on the catalyst surface that caused the catalyst deactivation. Moreover, the leaching rate of vanadium and tungsten, which are the main active components, increases and eventually lead to catalyst deactivation [53].

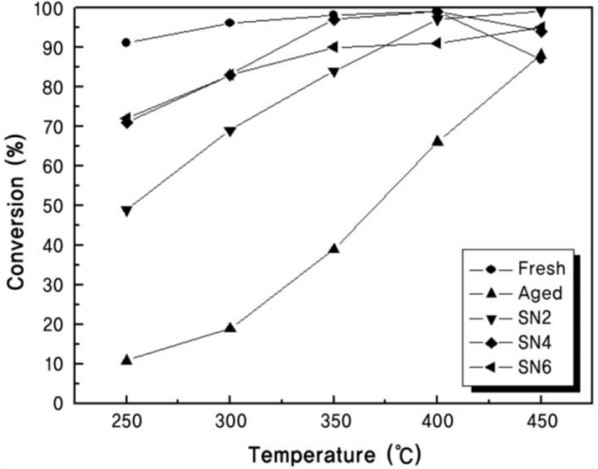

| Fresh | Commercial fresh catalyst |
|---|---|
| Aged | Commercial spent catalyst |
| SN2 | Commercial spent catalyst washed 2 M acid solution |
| SN4 | Commercial spent catalyst washed 4 M acid solution |
| SN6 | Commercial spent catalyst washed 6 M acid solution |

Acid solution : Mixed solution of sulfuric acid and nitric acid (mixed ratio = 1 : 1)

**Figure 25.** Conversion rate of NO over the prepared catalysts with sulfuric + nitric acid solution washing as a function of reaction temperature (space velocity = 6000 h$^{-1}$, NO/NH$_3$ ratio = 1:1, O$_2$ = 3.6 %($v/v$), NO = 300 ppm, SO$_2$ = 400 ppm) [53].

Despite many efforts being made to increase the life and stability of the SCR catalysts, they have a limited life depending on the operation conditions applied. The life of the catalyst used in the NH$_3$-SCR process is usually 3–5 years depending on the operating conditions of catalyst at each process site. In addition, only a few years ago, the activity of the SCR catalyst, which markedly deteriorated and reached the end of its life, was mostly treated by reclaiming the waste. Although research is being conducted worldwide to regenerate and use it, the currently studied regeneration method is to desorb the catalyst from the reactor and transport it to a regeneration plant to regenerate the degraded catalyst. In order to retain the original activity of the degraded catalyst, acid is used to remove the adsorbed coke and other particulate matters on the surface of the catalyst which was generally reported to be a time-consuming process [53].

### 3.3.2. Car Engines

Hydrocarbons, carbon monoxide and nitrogen oxides emitted by gasoline engines can be reduced by using three-way catalysts. Diesel engines also emit those reducing gases, although they need to be operated in excess oxygen environments. Due to the dominant concentration of reducing agents such as hydrocarbons and carbon monoxide in the exhaust gas, which actually need to be reduced to N$_2$ from NO$_x$, they reacted with excess oxygen. The diesel engine produces higher emissions of particle matter (PM) and NO$_x$ than gasoline engines [54]. In South Korea, diesel oxidation catalyst (DOC) and diesel particulate filter (DPF) were used to reduce the NO$_x$ as per the Euro-5 regulation. However, Euro-6 regulations, which applied from 2015, require a separate technology called SCR to reduce NO$_x$ as the emission volume of NO$_x$ is significantly strengthened, compared to Euro-5 (Figure 15). The reduction technique of NO$_x$ can be divided into two methods such as pre-processing technology (engine application technology, where both fuel and oxygen ratio adjust within engine) and post-processing technology (actual SCR catalytic technology) [54].

Among the post-processing technologies, the lean NO$_x$ trap (LNT) and SCR are the representative technologies that reduce nitrogen oxides. Urea-SCR is the most excellent device for removing NO$_x$ by spraying urea solution into the exhaust pipe to create ammonia by thermal hydrolysis (Figure 7). However, it should be considered that the system is complex and has problems such as low efficiency at low temperatures, the generation of solid byproducts and the use of low-grade elements, etc. These problems can be avoided by uniform mixing of the urea solution [55]. The driving characteristics of the vehicle cause a sudden change in engine load, resulting in different exhaust gas temperatures and concentrations of NO$_x$ (Figures 26 and 27) [56].

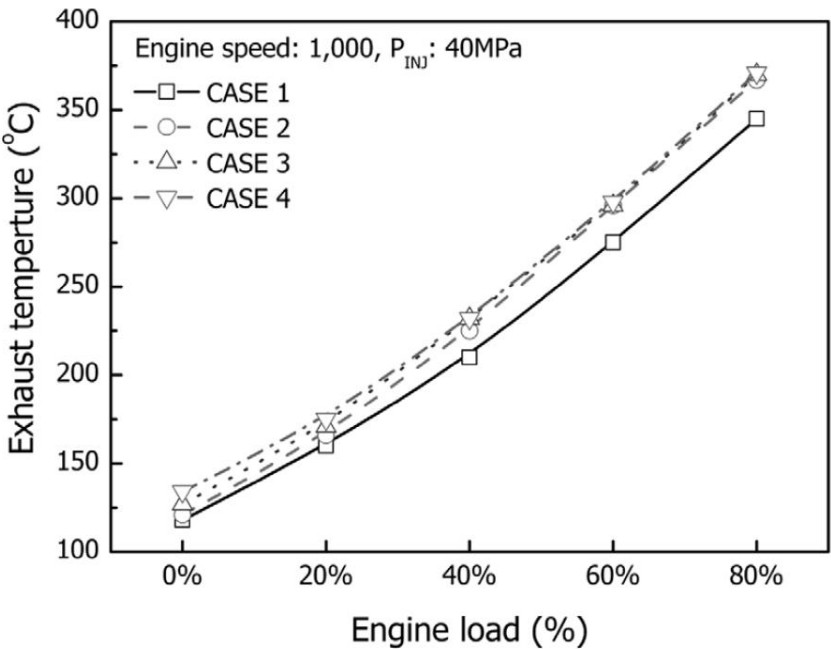

**Figure 26.** Comparisons of injection strategy of exhaust gas temperature [56].

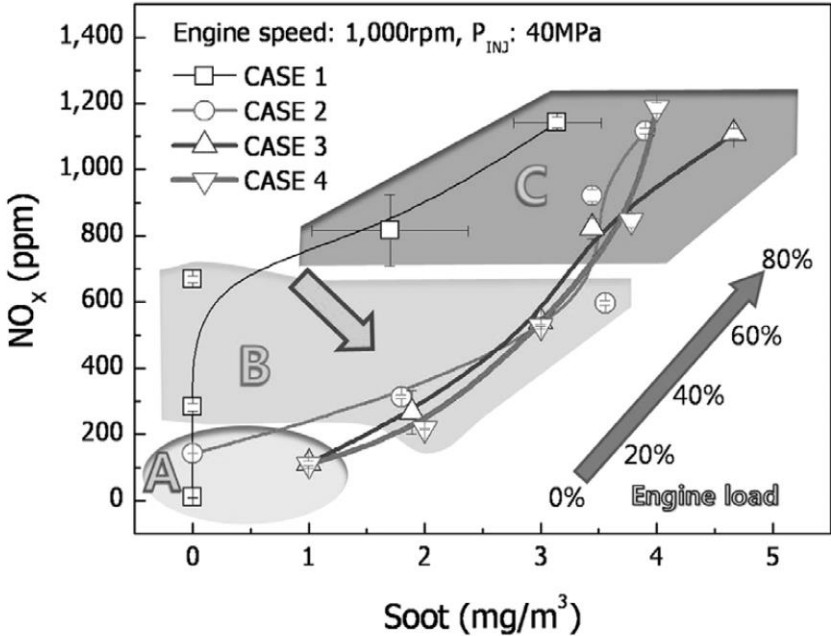

**Figure 27.** Effect of injection strategy on Soot and $NO_x$ emissions [56].

The most likely and effective deployment of the current vehicle SCR system is in the order of DOC and DPF followed by SCR. The SCR catalyst will also be exposed to high temperature exhausts during the forced regeneration of DPF PM combustion [57]. $V_2O_5$/WO/$TiO_2$ catalysts, widely used in plant industries, show excellent efficiency in the range of 300–400 °C, but the reaction activity and selectivity are significantly reduced at low temperature (below 200 °C) and high temperature (above 400 °C) [58]. The Cu-ZSM-5, a typical example of non-$V_2O_5$ based catalytic system, has also received an attention due to its fast rate of NO decomposition [59].

The vanadium catalyst generally used in the SCR process has a low deactivation at moderate reaction temperature even in the presence of sulfur, although the catalyst deactivated at high temperature. By contrast, the adsorption amount of ammonia is excellent on the zeolite catalysts in the wide reaction temperature range, but sulfur resistance is low. Han et al. investigated the performance

of the catalyst to overcome the shortcomings of vanadium catalyst and zeolite catalyst and maximize the catalyst life time as well as catalytic activity. They reported high temperature stability from the Fe-zeolite catalyst and sulfur resistance from the $V_2O_5$ catalytic system (Figure 28). Both the catalyst stability and sulfur resistance were enhanced when $V_2O_5$ was loaded into the Fe-zeolite catalyst (Figure 29) [60].

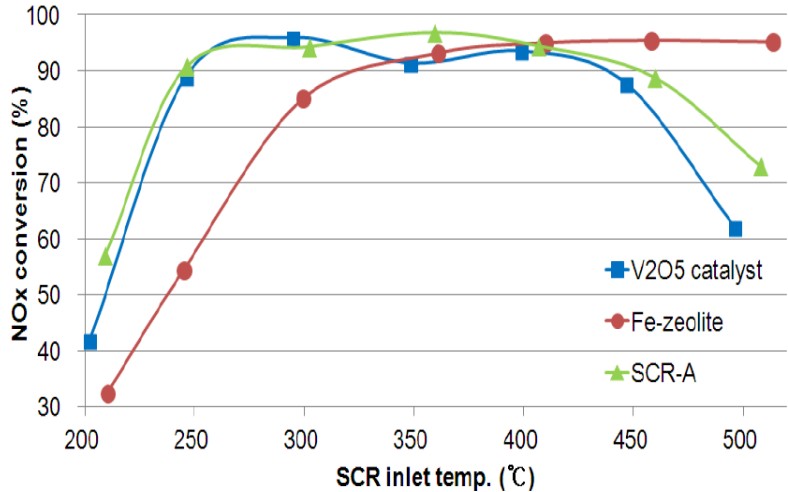

**Figure 28.** Comparison of DeNO$_x$ activity of SCR-A catalyst which was combined with $V_2O_5/TiO_2$ and Fe-zeolite [60].

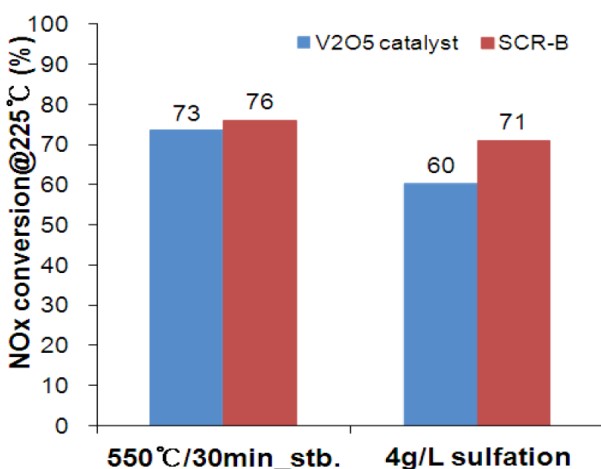

**Figure 29.** Steady-state engine test results at 225 °C before and after 4 g/L sulfation [60].

Huang et al. investigated the NH$_3$-SCR catalytic reaction on the zeolite catalyst with different structures. In more detail, using commercial zeolite catalyst (Clariant Co., Ltd. Daegu, Korea), they exchanged cations such as Cu and Fe to BEA (zeolite β) and MFI (ZSM-5) respectively. The BEA exchanged with Cu ions displayed a greater low temperature activity, and thus higher NO conversion rate than Fe exchanged on MFI. Furthermore, as shown Figure 30, NH$_3$ desorption peaks of BEA zeolite (B) was greatly influenced by the type of ion exchange metal ions rather than the structure of the zeolite's structure [61].

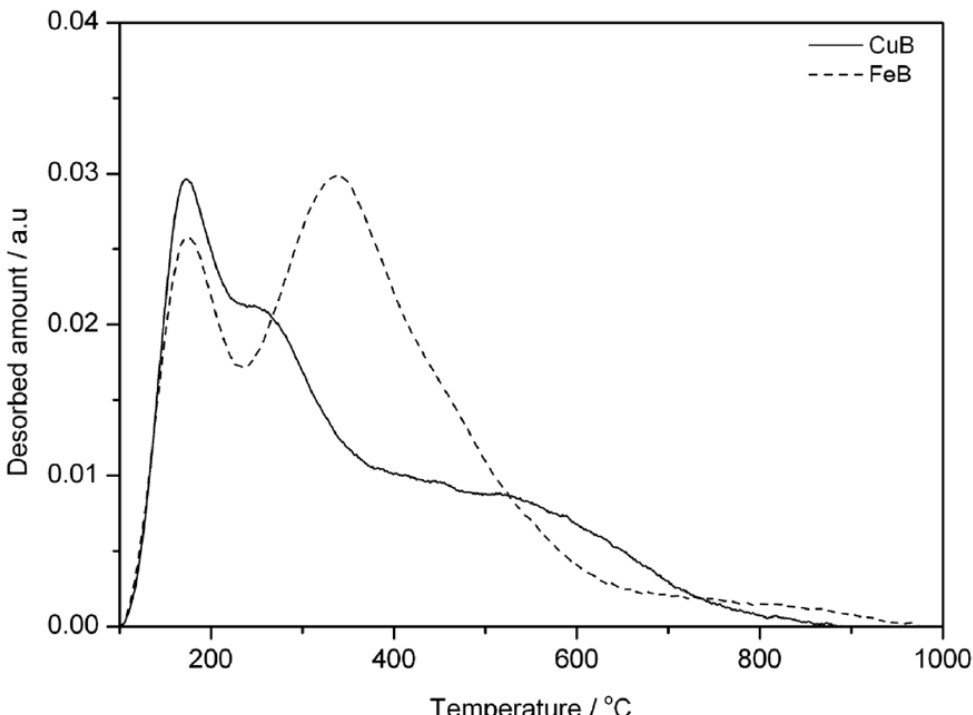

**Figure 30.** NH$_3$-TPD curves of CuB and FeB [61].

In another research, Seo studied the iron exchanged with Cu, Co, Ni, Mn, and Co in the zeolite, and investigated catalytic performance of DeNO$_x$. Also, the effects of the catalyst were investigated by adding some cations like Ba, K and Ca to the Cu-zeolite catalyst. The researcher reported that Cu and Mn showed a high NO$_x$ conversion at low reaction temperature of 200 °C (Figure 31). The Ba-modified zeolite showed a high activity at 500 °C, as compared with that of K- and Ca-modified catalysts, by inhibiting the agglomeration of Cu particles on the zeolite catalyst (Figure 32) [62].

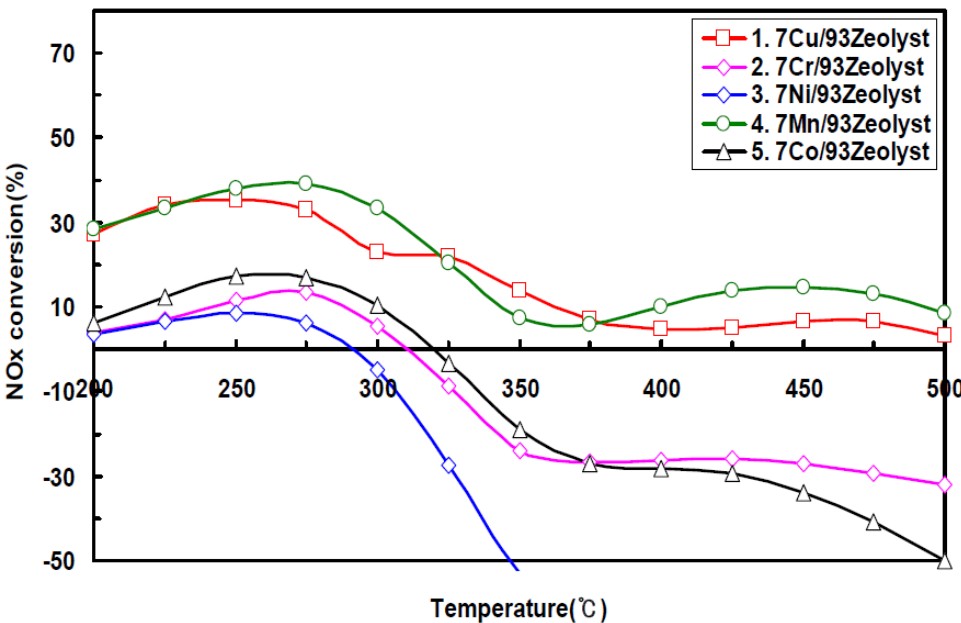

**Figure 31.** Conversion rate according to kind of transition metal [62].

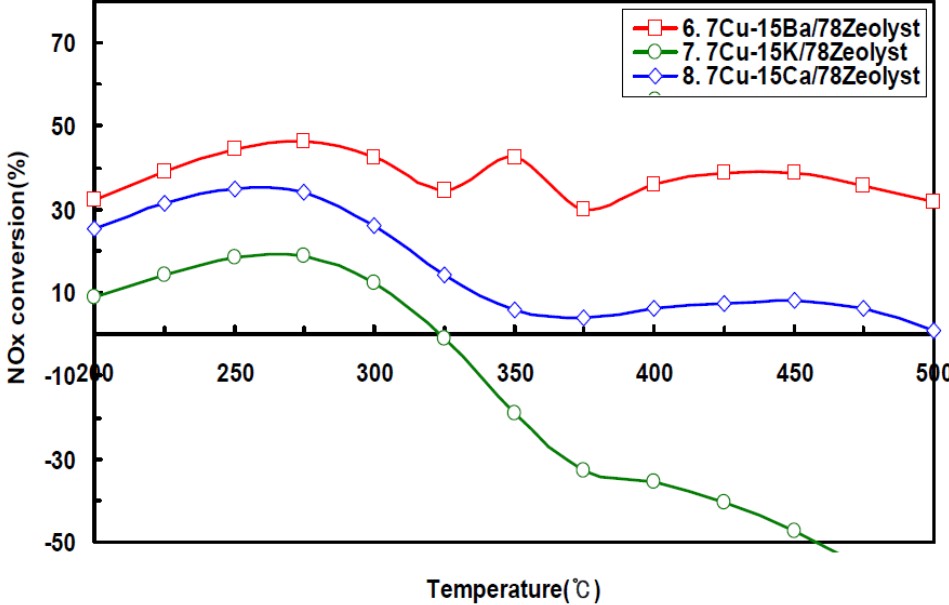

**Figure 32.** Conversion rate according to NO$_x$ storage material [62].

In the case of the direct ammonia-injection SCR system, the solid phase ammonia storage material is heated to generate ammonia gas, a reducing agent, and sprayed directly into the exhaust pipe in a gaseous phase. There is no discharge of solid byproducts, and the shape of the exhaust system required for uniform mixing of reducing agent is relatively free. In addition, since a series of processes such as thermal decomposition and hydrolysis are not required in low-temperature areas, which is advantageous in terms of reactivity and efficiency, this technique could be applied in the future [55].

The characteristics of NO$_x$ reduction rate, reaction rate, and ammonia slip of ammonia gas injection system were evaluated by Jung et al. via preparing ammonia gas injection system, which can replace the existing urea-SCR, using low temperature Cu-ZSM-5 catalyst. They reported that, unlike the urea solution, the response rate of ammonia gas was significantly improved due to the injection of reducing agents, with less stable adsorbed-phase occurring in the SCR catalyst (Figure 33) [63].

### 3.3.3. Marine Engines

The generation of nitrogen oxides and power supply for ships results from the internal combustion engines of ships. The exhaust emission characteristics of a heavy-duty diesel engine with real time analysis are shown in Figure 34 [64]. Since engine optimization alone does not satisfy Tier III's regulatory parameters on the reduction of nitrogen oxide emissions, the development of ship's exhaust gas post-processing device has become required to satisfy the enhanced emissions regulations of IMO [65].

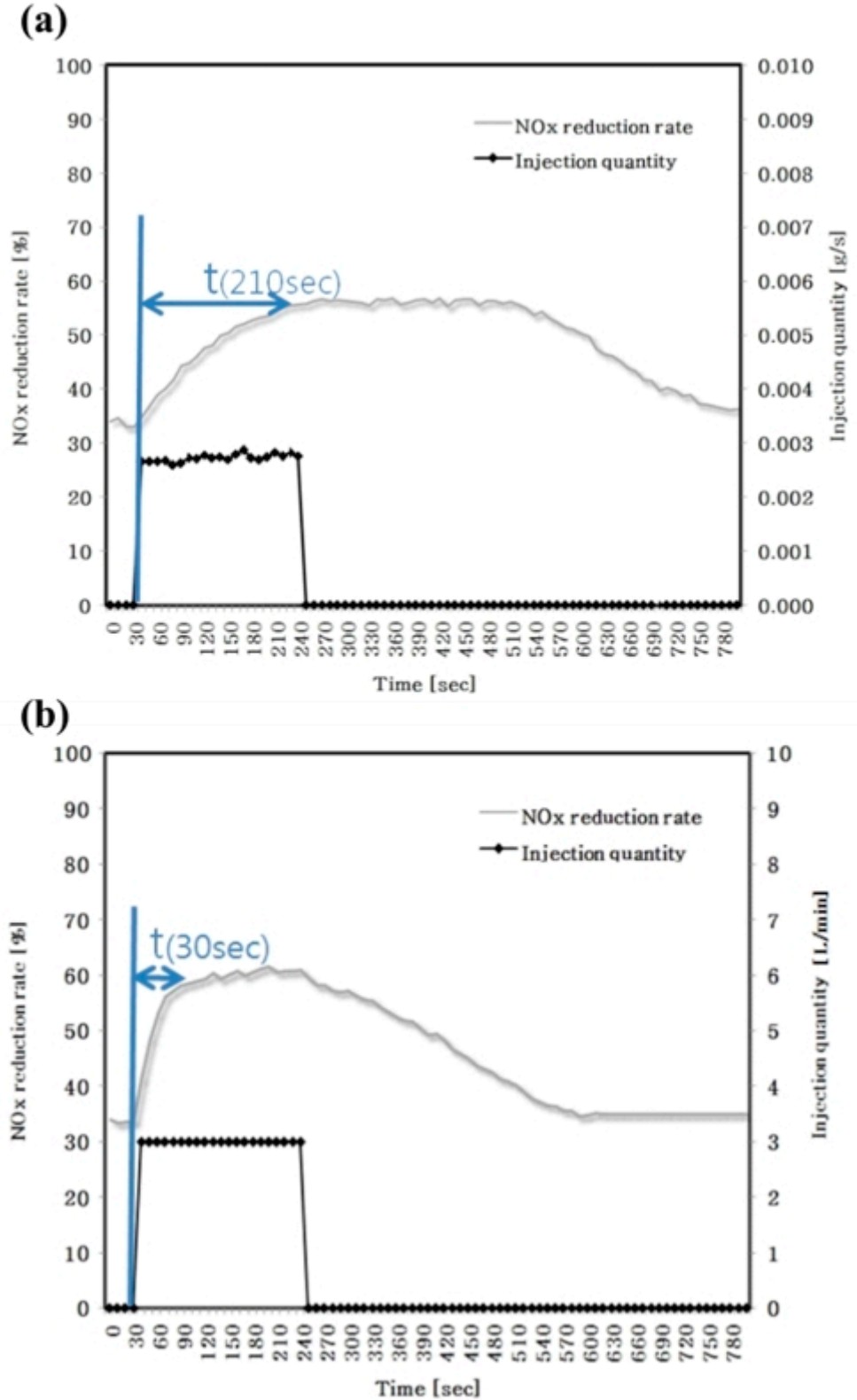

**Figure 33.** NO$_x$ reduction rate vs. time. Operating at 1500 rev/min, bmp 2 bar, (**a**) urea-SCR, (**b**) NH$_3$-SCR.

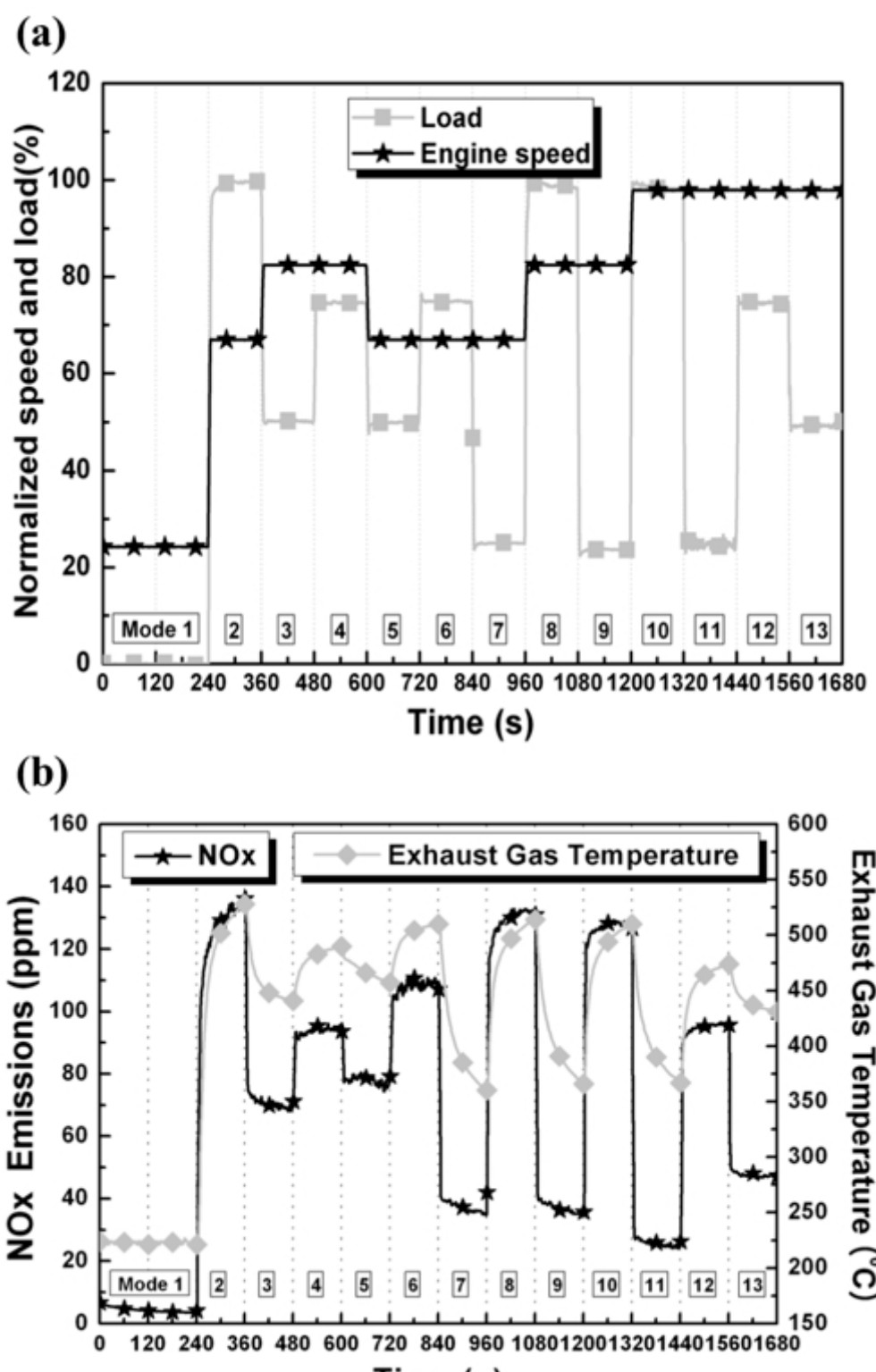

**Figure 34.** Exhaust emission characteristics of heavy-duty diesel engine: (**a**) real-time ECS (European Steady-State Cycle) test mode, (**b**) real-time $NO_x$ emission characteristics [64].

Most diesel engines for propulsion use a 2-stroke engine that is simple in structure and has good momentum. Diesel engines for power generation are mainly used with 4-stroke engines [66]. However, unlike automobile engines which have a temperature of 300 to 400 °C, the 2-stroke diesel engine for

ships only has a temperature of 200 to 300 °C, making it difficult for the SCR system to be applied to the system [65].

In addition, the catalyst used in the ship SCR process built within the ship, so the vibration resistance to withstand the internal engine, operation, and vibration caused by waves must be considered. Great importance should be given to the prevention of catalyst poisoning and the formation of ammonium bisulfate in the presence of $SO_2$ [67]. In order to solve this problem, various catalysts having nitrogen oxide reduction performance suitable for ship SCR process environment have been studied.

Kwon et al. studied the low temperature SCR reaction on $Mn/TiO_2$ catalyst using various commercial $TiO_2$ as supports (Figure 35). They reported that low $MnO_x$ surface density caused by high dispersion manganese oxides increased low-temperature SCR activity besides lowering the reduction temperature (from $MnO_2$ to $Mn_2O_3$) of surface MnOx on $TiO_2$ [68].

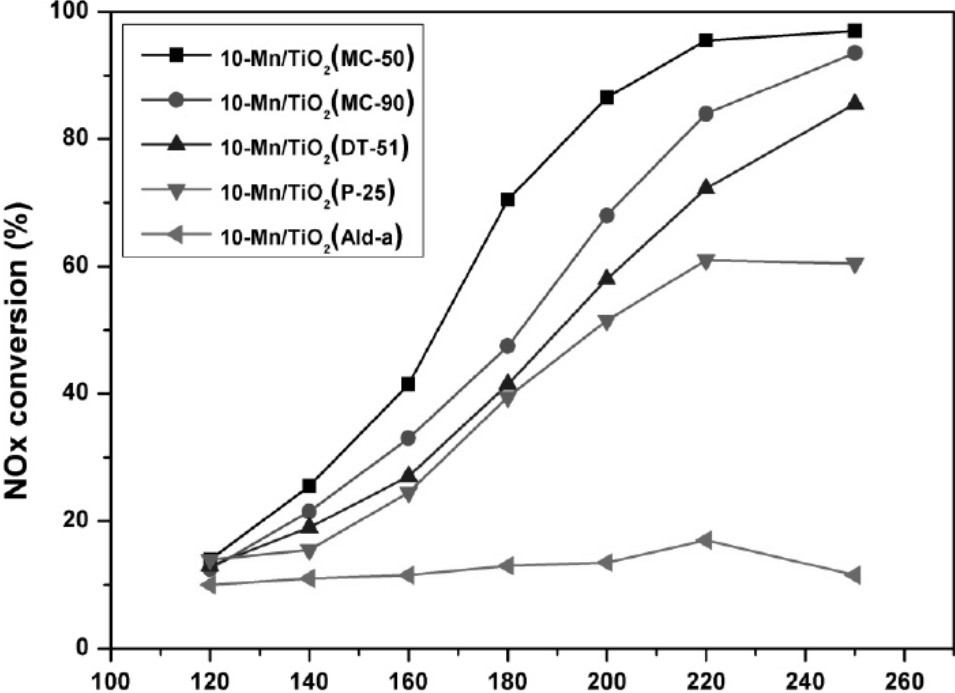

**Figure 35.** The effect of reaction temperature on $NO_x$ conversion over 10-$Mn/TiO_2$ catalysts ($NO_x$: 200 ppm, $NH_3/NO_x$: 1.0, $O_2$: 8%, S.V: 60,000 $h^{-1}$) [68].

As shown in Figure 36, Kang et al. investigated the $NO_x$ reduction rate using the single Cu and Mn oxide, and Cu-Mn mixed oxides catalyst. The presence of small amounts of copper oxide showed 100% $NO_x$ conversion compared with single metal oxide catalysts such as CuO and $MnO_x$ at wide temperature range (323–473 K) [69].

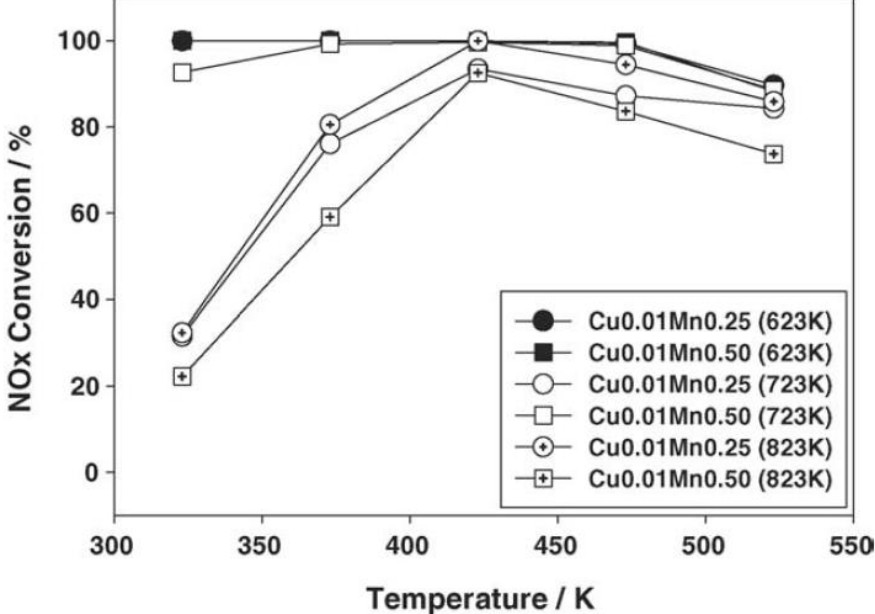

**Figure 36.** $NO_x$ conversions over Cu0.01Mn0.25 and Cu0.01Mn0.50 catalysts calcined at 623, 723 and 823 K. Reactants: 500 ppm NO, 500 ppm $NH_3$, 5 vol% $O_2$ in $N_2$, S.V: 30,000 $h^{-1}$. (Reprinted with permission form [69]. Copyright (2006) Elsevier).

Ship diesel fuel contains sulfur (Table 8), which will go through the combustion process and form $SO_2$, which is a known catalytic poisoning gas for the active species of the SCR catalysts. In more detail, $SO_2$ is the biggest problem for cold $NH_3$-SCR catalysts [70]. Many research efforts have been made to reduce the poisoning of SCR catalyst in the presence of sulfur.

**Table 8.** Properties of diesel fuel used in Korea naval vessels [71].

| Items | Standards | Results | Test Methods |
|---|---|---|---|
| Flash point (°C) | 40~ | 63.0 | KSM ISO 2719:2003 |
| Viscosity (40 °C, $mm^2$/s) | 1.5~6.0 | 3.301 | KSM ISO 3104:2008 |
| Sulfur (%) | ~1.0 | 0.028 | KSM ISO 8754:2003 |
| Cetane number | 40~ | 53.0 | KSM ISO 4264:2003 |
| Density@15 °C ($kg/m^3$) | 815–855 | 84.6 | KSM ISO 12185:2003 |

Ha et al. investigated the effects of moisture and $SO_2$ in gases (5% $H_2O$ and 100 ppm $SO_2$) on the selective catalytic recuperative activity of NO at low temperatures on Fe/zeolite catalyst. Additives such as Mn, Zr, and Ce were added into Fe/zeolite catalysts to check their catalytic efficacy. The NO conversion rate of Fe/BEA catalyst was reduced to 47% at 200 °C, which slightly increased to 53% on the MnFe/BEA in the presence of moisture and $SO_2$. The modified catalyst also displayed a greater stable activity than the Fe/BEA catalyst. Mn showed the effect of increasing the dispersion on Fe as compared to that of Zr and Ce, and also preventing the formation of a valuable aggregate of Fe [72].

Using the Cu-chabazite SCR catalyst, Nam investigated the effect of reducing nitrogen oxides by $SO_2$ poisoning at low temperatures between 150 and 300 °C. The researcher was able to measure $NO_x$ reduction for two hours at 150 °C, and then again at the same temperature for another two hours. Although the reduction rate of $NO_x$ was reduced by 50% compared to the new one, it was reported that the higher temperature near to 300 °C, the effect of sulfur poisoning was almost eliminated [73].

Lee et al. [74] investigated the effect of ceria loading over Sb-$V_2O_5$/$TiO_2$ catalyst for the selective catalytic reduction of $NO_x$ by $NH_3$. The $V_2O_5$-Ce/$TiO_2$ and Sb-$V_2O_5$-Ce/$TiO_2$ catalysts were prepared

by incipient wetness co-impregnation of vanadia and antimony on the pre-prepared $CeO_2/TiO_2$ support, which was synthesized by deposition precipitation method by hydrolysis with ammonium hydroxide. The addition of 10% ceria to $Sb-V_2O_5/TiO_2$ significantly enhanced the total acidity and redox properties, and thus the $NO_x$ conversion as well as $N_2$ selectivity (>95%) at wide temperature range of 220–500 °C (Figure 37). The X-ray diffraction (XRD) results indicated the active components of Sb and V were homogeneously dispersed over $CeO_2/TiO_2$. The results of NO and $SO_2$ TPD of 10% ceria-loaded $Sb-V_2O_5/TiO_2$ catalyst showed enhancement of NO adsorption and $SO_2$ inhibition properties, which is thought to play a significant role in the long-term stability of the catalyst during an $SO_2$ on–off study for 38 h at 240 °C.

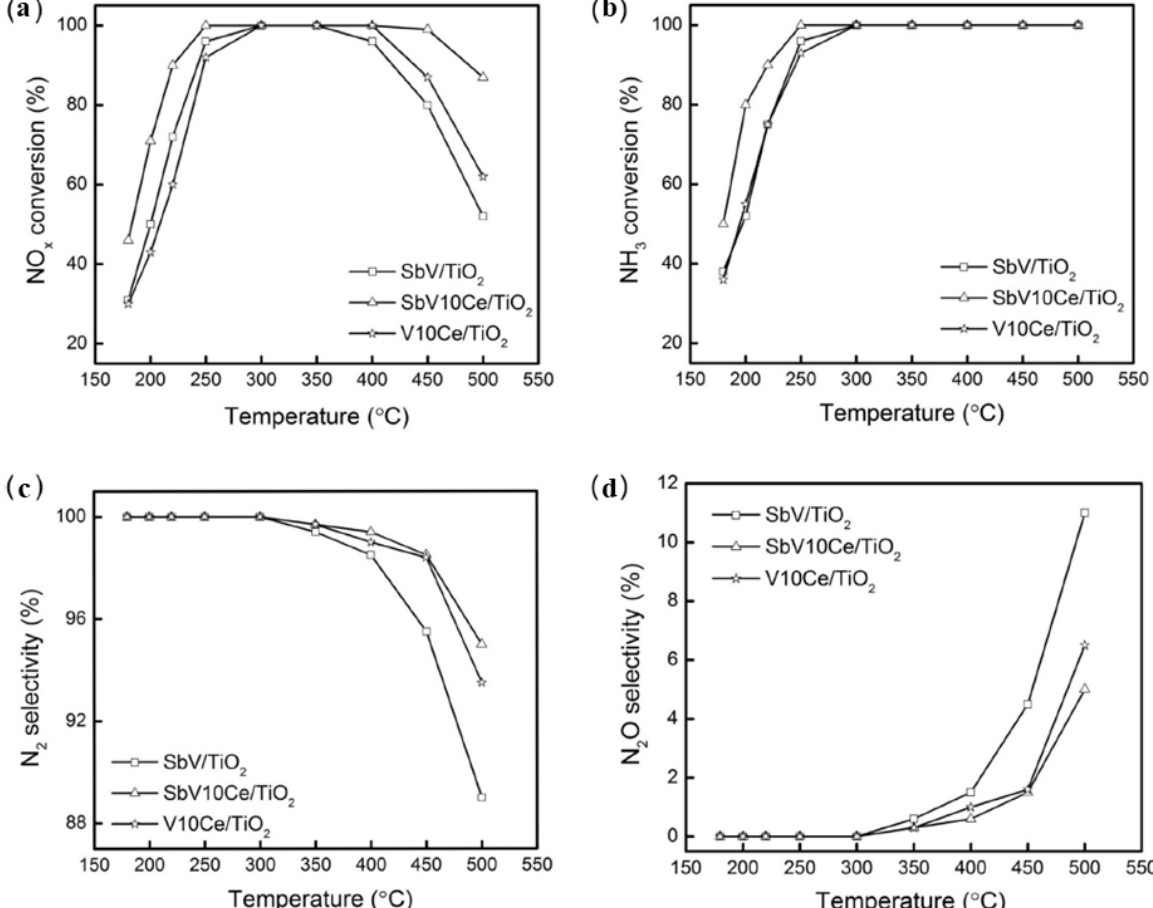

**Figure 37.** Activity of $SbV/TiO_2$, $SbV10Ce/TiO_2$ and $V10Ce/TiO_2$ samples calcined at 500 °C under $H_2O$ and $SO_2$ (**A**) $NO_x$ conversion (**B**) $NH_3$ conversion (**C**) $N_2$ selectivity and (**D**) $N_2O$ selectivity. (Reaction conditions: $[NO_x]$ = $[NH_3]$ = 800 ppm, $[O_2]$ = 3 vol%, $[H_2O]$ = 6 vol%, $[SO_2]$ = 800 ppm, $N_2$ balance, S.V = 60,000 $h^{-1}$). (Reprinted with permission form [74]. Copyright (2013) Elsevier).

To investigate the effect of pore structure of $TiO_2$ on sulfur poisoning, Youn et al. studied 5 wt% $V_2O_5$ supported on the two types of $TiO_2$ (VT) having distinctive pore, mesopore (DT-51) and micropore (microporous $TiO_2$) structures for selective catalytic reduction of $NO_x$ with $NH_3$ ($NH_3$ SCR). The catalyst prepared by the wet impregnation method of vanadium solution on $TiO_2$. During the SCR reaction in the presence of $SO_2$ for 12 h, 5 wt% VT (DT-51) showed more drastic decrease in activity than 5 wt% VT (micro). They reported that a larger amount of $SO_2$ was desorbed over the DT-51 catalyst used during the temperature-programmed decomposition with the elemental analysis. The 5 wt% VT (micro) having V=O bonds that showed stronger sulfur resistance, but 5 wt% VT (DT-51) having V–O–V bonds that lost their activity more severely than the former. They also summarized

that the different vanadium species on the pore structure of $TiO_2$ have a significant effect on the sulfur poisoning during SCR reaction [75].

## 4. Conclusions

In this paper, we reviewed the results of the selective catalytic reduction (SCR) process to remove nitrogen oxide ($NO_x$), an air pollutant, in South Korea. Plant industry, car engines and marine engines are the main $NO_x$ producers. The technologies used to reduce the $NO_x$ emission are summarized in the following section.

Power plants: in the case of power plants, steady operating performance is required among SCR catalytic performances. Coal-fired power plants generate the high-temperature along with exhaust gases, which may deactivate and inhibit the stable operation of the SCR catalyst. Therefore, the SCR technology is focused on the identification of the direction of increasing the catalyst stability at high temperature operation for large-scale equipment. In South Korea, power plants under KEPCO account for about 81% SCR catalytic system for fixed pollutants. Therefore, SCR catalyst research on fixed pollutants has been conducted in response to the requirements of the power plant. On the other hand, thermal power plant is responsible for 69% of power generation, of which about 46% is fueled by coal. Therefore, SCR catalysts for fixed pollutants are being studied to reflect the characteristics of coal-fired power plants. As of 2015, the regulation of nitrogen oxide emission for these fixed sources has been greatly strengthened, so it is required to increase the reduction rate of nitrogen oxides in SCR units. In recent years, research has been conducted to improve the economy and reduce waste of power plants by regenerating large amounts of deactivated catalysts obtained from thermal power plants.

Car engines: automobile engines are divided into gasoline engines and diesel engines. There are differences in the composition of air pollutants (gasoline: hydrocarbons, carbon monoxide, nitrogen oxides/diesel: oxygen, nitrogen oxides, oxidizing substances), which are emitted from them. Three-way catalytic gasoline engines (number of SCR units; lower oxygen conditions rarely used) and diesel engines (exist always oxygen) are two separate device technologies. In particular, in the case of diesel engine, the generation of $NO_x$ is high due to the combustion characteristic of excessive oxidizing agent, and thus, SCR technology for automobiles is mainly focused and researched on diesel engine. In South Korea's automobile industry, the regulation of nitrogen oxides from diesel engines is adopted from European emission regulations, and the introduction of these regulations has led to the need for automotive SCR devices. The emission regulation is applied regardless of the load of the diesel engine, so the ability to remove $NO_x$ in the low-temperature zone, which is the initial starting point of the vehicle, is required. In addition, due to particle matter (PM) regulation, the diesel particle filter (DPF) is mounted on the vehicle in which high-temperature exhaust gases are generated. Therefore, SCR catalytic technology is suitable to be installed after the DPF device so that the catalytic system can remove $NO_x$ in the low-temperature region and also resist the heat generated during the high temperature exhaust gas. In this regard, automotive SCR catalysts have also been studied in response to this demand. In addition, the research on SCR catalysts by direct injection of ammonia is required in the future, which may provide a high performance $NO_x$ removal, according to the Real Driving Emission (RDE) measurement method.

Marine engines: SCR denitrification catalysts are generally used in marine engines. The exhaust gas temperature of ship engine is below 300 °C, so the catalyst should be active in the low-temperature range. The proportion of international ships arrival or passing through the area of seas in between countries contribute larger emissions of $NO_x$ than Korean domestic ships. The capacity of international ships is also very high as compared to domestic ones and, therefore, they emit higher amounts of $NO_x$. These ships are required to comply with international regulations centered on IMO rather than domestic regulations. Since 2016, the IMO's regulation of nitrogen oxides has been greatly strengthened to Tier III, which requires the introduction of additional SCR facilities for exhaust gas to reduce nitrogen oxides through optimization of existing combustion conditions and engine technology. Accordingly, domestic SCR catalysts for ships have also been revived to meet these demands, and research on

SCR catalysts for ship engines is currently underway. In particular, the fuel for ships is high in sulfur content and exhaust gases are generated even at low-temperature due to low load operation. Therefore, SCR catalysts targeting domestic ships are also being studied for catalysts having activity against low temperature SCR reaction and poison resistance to sulfur.

Many efforts have been made to develop South Korea's own technology to reduce the amount of nitrogen oxide. We hope that the SCR technology developed will definitely have a positive impact on air pollution.

**Author Contributions:** H.-S.K., S.K. and J.K. jointly collected the literatures and equally contributed for this review paper. S.-H.K. and J.-H.K. help for some data collection. J.-H.R. and J.-W.B. revised the draft and supervised the work. All authors have read and agreed to the published version of the manuscript.

**Funding:** This work was supported by Korea Institute of Energy Technology Evaluation and Planning (KETEP) under the "Energy Efficiency and Resources Programs" (Project No. 20181110100320) of the Ministry of Knowledge Economy, Republic of Korea.

**Conflicts of Interest:** The authors declare no conflict of interest.

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
