# Peer review of "Current Catalyst Technology of Selective Catalytic Reduction (SCR) for NOx Removal in South Korea"

_catalysts, doi:10.3390/catal10010052_

Round 1
Reviewer 1 Report
Right on these days we are celebrating a global conference against climate change. Therefore, any research paper like the one here presented can be considered a contribution that deserves consideration. The authors have performed a nice review of the SCR technologies in their own country, South Korea. It is well known that these technologies are one of the main solutions to deal with the environmental pollution due to NOx. In general the review is well structured and complete as it covers the main sources of this pollutant and the corresponding strategies, not only the current ones but also the prospectives for the future. It also handles an interesting list of bibliographic references, many of them being quite recent. However, there are several limitations that must be overcome before it can be published. The major one is English, which should be thoroughly revised both in terms of grammar and syntax. Some examples (as a tip to help the authors) can be found in lines: 46-47, 82-85, 90-91, 93-95, 102, 115, 122, 142-144, 162, 164, 168, 170-174, 178, 188, 190, 192-194, 197-198, etc, etc. The whole list would be very long. Sometimes the sentences can be hardly understood, others there is wrong use of verbs, or inappropraite mix of tenses (future, present and past), or lack of articles. Please, look for the help of a native speaker.
The second problem is related with Figures and Tables. On one hand, there is an incorrect numeration of them from a certain point of the manuscript what denotes a lack of care. In page 15 , line 55, Figure 1 must be Figure 16; in page 16, line 64, Figure 2 must be Figure 17; in page 18, line 109, Figure 3 must be Figure 18; in page 19, lines 120, 127 and 130, Figures 4, 5 and 1 should be Figures 19, 21 and 20, respectively (by the way, the latter two exchanged in order); in page 20, line 143, Figure 6 should be Figure 22; in page 21, line 159, Figure 7 must be Figure 23; in page 23, line 213, Figure 8 must be Figure 26; in page 25, line 231, Figure 9 must be Figure 28; in page 26, line 244, Figure 10 must be Figure 30; and in page 30, line 335, Figure 11 is indeed Figure 37. On the other hand, some figures and tables are not cited in the text: Table 4, Table 7, Figure 25, Figure 34 and Figure 35.
The third main deficiency is that the letter size is too small and thus not readable in some figures and tables. For example in Table 2, Figure 14 and Figure 33. Also, Tables and Figures must be always self-explaining. In this sense: Figure 1 caption misses the units (μg/m3); Table 2 should have a footnote indicating the meaning of SCC; Figure 12 should include an explanation of the number code, and an indication in its caption saying that data are for South Korea; Figure 13 misses the units (ppm) in the y-axis; and Tables 5 and 6 headings should indicate that the information provided is for South Korea.
In addition:
1) In line 206 of page 9, the authors refer to "very recent reviews", but 2 of the 3 ones that are mentioned are from 2012, seven years ago so far!
2) In page 19, line 126, the mentioned temperature should be 600 ºC instead of 700 ºC.
3) Reference 17 misses the title.
Author Response
Reply to reviewer(#1)’s comments:
Right on these days we are celebrating a global conference against climate change. Therefore, any research paper like the one here presented can be considered a contribution that deserves consideration. The authors have performed a nice review of the SCR technologies in their own country, South Korea. It is well known that these technologies are one of the main solutions to deal with the environmental pollution due to NOx. In general the review is well structured and complete as it covers the main sources of this pollutant and the corresponding strategies, not only the current ones but also the prospectives for the future. It also handles an interesting list of bibliographic references, many of them being quite recent. However, there are several limitations that must be overcome before it can be published. The major one is English, which should be thoroughly revised both in terms of grammar and syntax.
--> We are sincerely thankful to reviewer’s comments and valuable suggestions that helped us to improve the review paper with better presentation. All the corrections and modifications have been highlighted by yellow color.
Some examples (as a tip to help the authors) can be found in lines: 46-47, 82-85, 90-91, 93-95, 102, 115, 122, 142-144, 162, 164, 168, 170-174, 178, 188, 190, 192-194, 197-198, etc, etc. The whole list would be very long. Sometimes the sentences can be hardly understood, others there is wrong use of verbs, or inappropraite mix of tenses (future, present and past), or lack of articles. Please, look for the help of a native speaker.
--> The whole manuscript has been thoroughly revised for grammatical errors, verbs and tenses, and checked and rearranged the sentence formation by experts as possible as we can. All the corrections and modifications have been highlighted by yellow color. The line numbers in the revised draft are changed due to an additional corrections as per the reviewer’s comments.
The second problem is related with Figures and Tables. On one hand, there is an incorrect numeration of them from a certain point of the manuscript what denotes a lack of care. In page 15 , line 55, Figure 1 must be Figure 16; in page 16, line 64, Figure 2 must be Figure 17; in page 18, line 109, Figure 3 must be Figure 18; in page 19, lines 120, 127 and 130, Figures 4, 5 and 1 should be Figures 19, 21 and 20, respectively (by the way, the latter two exchanged in order); in page 20, line 143, Figure 6 should be Figure 22; in page 21, line 159, Figure 7 must be Figure 23; in page 23, line 213, Figure 8 must be Figure 26; in page 25, line 231, Figure 9 must be Figure 28; in page 26, line 244, Figure 10 must be Figure 30; and in page 30, line 335, Figure 11 is indeed Figure 37. On the other hand, some figures and tables are not cited in the text: Table 4, Table 7, Figure 25, Figure 34 and Figure 35.
--> Thank you for the reviewer's comment. There seems to be an error when uploading the final version of the review paper. This part has been revised in the manuscript. Some non-tagged tables and figures are corrected and mentioned in the text.
The third main deficiency is that the letter size is too small and thus not readable in some figures and tables. For example in Table 2, Figure 14 and Figure 33. Also, Tables and Figures must be always self-explaining. In this sense: Figure 1 caption misses the units (μg/m3); Table 2 should have a footnote indicating the meaning of SCC; Figure 12 should include an explanation of the number code, and an indication in its caption saying that data are for South Korea; Figure 13 misses the units (ppm) in the y-axis; and Tables 5 and 6 headings should indicate that the information provided is for South Korea.
--> There are some poor quality figures in the published papers. We have been tried our level best to increase the quality of the figures and tables wherever necessary. In addition, the missing unit in Fig. 13 is included in the text. Korea's national data only given in the Table 5 and 6.
In addition:
1) In line 206 of page 9, the authors refer to "very recent reviews", but 2 of the 3 ones that are mentioned are from 2012, seven years ago so far!
--> Yes, we agree with the reviewer’s comment. We now changed the sentence as follows:
The details of the active metals, and their supports and degree of deactivation rate for H2-SCR were already covered in the past review papers.
2) In page 19, line 126, the mentioned temperature should be 600 ºC instead of 700 ºC.
--> Thank you for the reviewer's comment. The correct temperature value given in the revised manuscript.
3) Reference 17 misses the title.
--> Thank you for a good point. Title has been included in accordance with the journal’s in-house style.

Reviewer 2 Report
The manuscript study Current Catalyst Technology of Selective Catalytic Reduction (SCR) for NOx Removal in South Korea. In any case, the results are interesting and it could be published after some modifications.
The format of the article should be reviewed carefully because it is full of mistakes. Here are some examples:
Page 20. Figure 20. The figure is of poor quality. In addition, it is difficult to obtain information from this figure. Page 15. Figure 14 (a, b). The signs on the figures don't read well, they should be bigger. Page 17. Figure 17. The signs on the figures don't read well, they should be bigger. Figure 29 is very large in relation to the other figures. You have to check the size of all the figures.
In table 5 the decimal digits are given with commas. In table 6 with periods...It must be consolidated throughout the text.
In some tables the text is centered and in other tables it is not. It must be consolidated throughout the text.
Author Response
Reply to reviewer(#2)’s comments:
The manuscript study Current Catalyst Technology of Selective Catalytic Reduction (SCR) for NOx Removal in South Korea. In any case, the results are interesting and it could be published after some modifications.
--> We are sincerely thankful to reviewer’s comments and valuable suggestions that helped us to improve the review paper with better presentation. All the corrections and modifications have been highlighted by yellow color.
The format of the article should be reviewed carefully because it is full of mistakes. Here are some examples:
Page 20. Figure 20. The figure is of poor quality. In addition, it is difficult to obtain information from this figure. Page 15. Figure 14 (a, b). The signs on the figures don't read well, they should be bigger. Page 17. Figure 17. The signs on the figures don't read well, they should be bigger. Figure 29 is very large in relation to the other figures. You have to check the size of all the figures.
--> There are some poor quality figures in the published papers. We have been tried our level best to increase the quality of the figures and their signs and tables wherever necessary. In addition, we reduced the size of Fig. 29.
In table 5 the decimal digits are given with commas. In table 6 with periods...It must be consolidated throughout the text.
--> The values are given with commas in Table 5 are correct as the unit is different in Table 5 than the values given in Table 6.
In some tables the text is centered and in other tables it is not. It must be consolidated throughout the text.
--> The text contained in the table is centered uniformly.

Reviewer 3 Report
The manuscript presents a complete review on the SCR technology specifically addressed to the air pollution control in South Korea.
The Authors correctly described the SCR process for fixed and mobile applications with reference to emissions limits imposed in South Korea.
What it is not clear is if the experimental results reported from the literature are directly related to the materials applied in south Korea or not. The Authors should clarify this point.
Author Response
Reply to reviewer(#3)’s comments:
The manuscript presents a complete review on the SCR technology specifically addressed to the air pollution control in South Korea.
The Authors correctly described the SCR process for fixed and mobile applications with reference to emissions limits imposed in South Korea.
--> We are sincerely thankful to reviewer’s comments and valuable suggestions that helped us to improve the review paper with better presentation. All the corrections and modifications have been highlighted by yellow color.
What it is not clear is if the experimental results reported from the literature are directly related to the materials applied in south Korea or not. The Authors should clarify this point.
--> The reported literature results are directly related to the materials applied only in South Korea. We have clearly mentioned this details in the whole manuscript. For instance, we present the following sentences in both abstract and conclusions:
Abstract
In this paper, the results of recent catalyst studies on NOx removal by selective catalytic reduction are reviewed with the sources and the regulations applied according to the national characteristics of South Korea. Specifically, we emphasized the three major NOx emissions sources in South Korea such as plant, automobile, and ship industries and their catalyst technologies used.
Conclusions
We reviewed the results of the selective catalytic reduction (SCR) process to remove nitrogen oxide (NOx), an air pollutant, in South Korea. Plant industry, car engines and marine engines are the main sources of NOx producers. The technologies used to reduce the NOx emission are summarized in the following section.

Round 2
Reviewer 1 Report
In general the authors have implemented the suggested corrections. However, there are still deficiencies in the English that must be overcome before this review can be published. Some of them are new and appeared when they wrote new parts in the text. Sorry, but I cannot believe that the manuscript was revised by a native speaker because I am not a native speaker either and I was able to detect the following errors without reading the whole text (so there must be even more):
Line 144: adopted instead of adopt.
Line 239: regions instead of region
page 13, line 15: has fixed instead of is fixed
page 13, line 17: strengthen and not strengthen in
page 14, line 31: contributors to the non-road pollution
page 18, line 82-85: Four layers SCR catalytic system also displayed a good efficiency for removal of NOx. However, they suffered a pressure drop, due to additional caltalytic layers, which increased not only the overall cost of the DeNOx process but also the ammonia distribution in the nozzle of ammonia injection facility.
page 18, line 100: any catalyst
page 19, lines 199-121: ...that at temperatures above 400 ºC the phase transition of V2O5 occurred and condensation caused the decrease of the specific surface area (Figure 19) and hence of the catalytic activity.
Figures 20 and 21 captions: at different calcination temperatures
Page 21, line 139: loaded
page 21, line 141: 900 h
page 21, lines 146 and 148: may be
page 21, line 149: this salt also allowed covering
page 21, line 151, page 26, line 223, page 30, line 289, page 32, line 309, page 33, line 344: in the presence of
page 26, line 229: was loaded
page 33, line 340: studied instead of studied with
page 33, line 346: cancel "Since", and showed (not show)
page 34, line 383: is mounted
page 34, line 384: suitable to be installed
page 34, line 390: is generally used
page 34, lines 393-394: are also very high... ones....and therefore they emit higher amounts of NOx.
Author Response
Reply to reviewer(#1)’s comments:
In general, the authors have implemented the suggested corrections. However, there are still deficiencies in the English that must be overcome before this review can be published. Some of them are new and appeared when they wrote new parts in the text. Sorry, but I cannot believe that the manuscript was revised by a native speaker because I am not a native speaker either and I was able to detect the following errors without reading the whole text (so there must be even more):
Line 144: adopted instead of adopt.
Line 239: regions instead of region
page 13, line 15: has fixed instead of is fixed
page 13, line 17: strengthen and not strengthen in
page 14, line 31: contributors to the non-road pollution
page 18, line 82-85: Four layers SCR catalytic system also displayed a good efficiency for removal of NOx. However, they suffered a pressure drop, due to additional caltalytic layers, which increased not only the overall cost of the DeNOx process but also the ammonia distribution in the nozzle of ammonia injection facility.
page 18, line 100: any catalyst
page 19, lines 199-121: ...that at temperatures above 400 ºC the phase transition of V2O5 occurred and condensation caused the decrease of the specific surface area (Figure 19) and hence of the catalytic activity.
Figures 20 and 21 captions: at different calcination temperatures
Page 21, line 139: loaded
page 21, line 141: 900 h
page 21, lines 146 and 148: may be
page 21, line 149: this salt also allowed covering
page 21, line 151, page 26, line 223, page 30, line 289, page 32, line 309, page 33, line 344: in the presence of
page 26, line 229: was loaded
page 33, line 340: studied instead of studied with
page 33, line 346: cancel "Since", and showed (not show)
page 34, line 383: is mounted
page 34, line 384: suitable to be installed
page 34, line 390: is generally used
page 34, lines 393-394: are also very high... ones....and therefore they emit higher amounts of NOx.
--> We appreciate the detailed reviewer comments. The revision was completed by reflecting the correct opinion. Thanks to you, the completeness of our paper has increased. Thank you again.
Reviewer 2 Report
The authors have made the changes and the article has been substantially improved.
Author Response
We appreciate the detailed reviewer comments. The revision was completed by reflecting the correct opinion. Thanks to you, the completeness of our paper has increased. Thank you again.